# A Trichotomy for List Transductive Online Learning

**Steve Hanneke** [1]   **Amirreza Shaeiri** [1]

## Abstract

List learning is an important topic in both theoretical and empirical machine learning research, playing a key role in the recent breakthrough result of (Brukhim et al., 2022) on the characterization of multiclass PAC learnability, as well as the ambiguity of labels in computer vision classification tasks, among others. In this paper, we study the problem of list transductive online learning. In this framework, the learner outputs a list of multiple labels for each instance rather than just one, as in traditional multiclass classification. In the realizable setting, we demonstrate a trichotomy of possible rates of the minimax number of mistakes. In particular, if the learner plays for $T \in \mathbb{N}$ rounds, its minimax number of mistakes can only be of the orders $\Theta(T)$, $\Theta(\log T)$, or $\Theta(1)$. This resolves an open question raised by (Hanneke et al., 2024b). On the other hand, in the agnostic setting, we characterize the learnability by constructively proving the $\widetilde{\mathcal{O}}(\sqrt{T})$ upper bound on the minimax expected regret. Along this way, we also answer another open question asked by (Moran et al., 2023). To establish these results, we introduce two new combinatorial complexity dimensions, called the Level-constrained $(L + 1)$-Littlestone dimension and Level-constrained $(L + 1)$-Branching dimension, if the list size is $L \in \mathbb{N}$. Eventually, we conclude our work by raising an open question regarding eliminating the factor of list size, which seems to be a crucial step, as it has consistently appeared in previous works on this subject.

## 1. Introduction

List learning is a significant subject in both theoretical and empirical machine learning research. From a theoretical point of view, a key technique in the recent breakthrough

result by (Brukhim et al., 2022) on the combinatorial characterization of multiclass PAC learnability is the use of list learners. Subsequent works, including (Charikar & Pabbaraju, 2023), (Moran et al., 2023), (Brukhim et al., 2024), (Hanneke et al., 2024a), and (Pabbaraju & Sarmasarkar, 2024) have studied the notion of list learnability in various contexts, such as PAC learning, online learning, boosting, as well as sample compression and uniform convergence, and real-valued regression, accordingly. Moreover, this topic brings to mind the fundamental setting of list-decodable learning in statistics. For a detailed discussion, for example, refer to Chapter 5 of the recent textbook by (Diakonikolas & Kane, 2023). From an empirical point of view, there are many scenarios in which one may prefer the list learning approach to the classical multiclass classification. For instance, in recommendation systems, the objective is often to present a short list of items to users, trying to ensure that the user will select one of the items from the list. As another example, in computer vision classification tasks, predicting a list of labels can potentially prevent label ambiguity. Furthermore, this subject may bring to mind the fundamental setting of conformal prediction in practical applications, which can be viewed as a dual counterpart of the list setting.

Consider a round-robin tennis tournament consisting of $T \in \mathbb{N}$ matches, all scheduled in advance. In tennis, each set continues until one player leads by at least two games, so sets can, in theory, play on indefinitely; this means the space of possible exact outcomes is countably infinite. A gambler, aware of the players participating in each match based on the schedule, aims to predict the exact outcomes. Since predicting the precise results is challenging, the gambler is allowed to submit a list of $L \in \mathbb{N}$ possible outcomes for each match before it begins. After each match concludes, the actual result is disclosed. The gambler earns a profit if the actual outcome is among the predictions they have submitted for that match. Given minimal assumptions about the nature of the matches, how can the gambler effectively select predictions to maximize the number of rounds that yield a profit?

The aforementioned example, along with many other real-world scenarios involving possibly adversarially chosen pre-specified schedules, can be formulated within the framework called *List Transductive Online Learning*. This framework is informally defined as follows. Initially, an adversary

---

[1]Department of Computer Science, Purdue University, West Lafayette, IN, USA. Correspondence to: Steve Hanneke <steve.hanneke@gmail.com>.

*Proceedings of the 42$^{nd}$ International Conference on Machine Learning*, Vancouver, Canada. PMLR 267, 2025. Copyright 2025 by the author(s).

selects a finite sequence of instances, such as images, of which the learner is aware. In each subsequent round, the learner must predict a list of labels, such as a list of possible categories for an image, for the next instance in the sequence. Following each prediction, an adversary reveals the correct label. Importantly, in this setting, the learner provides a list of multiple labels for each instance instead of a single label, as in traditional multiclass classification. Moreover, the primary quantity of interest in this framework is the notion of the number of mistakes the learner makes over time. In particular, a mistake occurs if the correct label is not included in the learner's predicted list of labels. For simplicity, the informal formulation is presented here in the context of deterministic learners.

To derive meaningful results, this work adopts the well-established notion of a concept class, consisting of functions mapping the instance space to the label space. In the realizable setting, we assume that the sequence generated by the adversary is consistent with at least one of the concepts within the concept class. Furthermore, in this setting, we focus specifically on minimizing the number of mistakes made by the learner as the primary objective. In contrast, in the agnostic setting, no assumptions are made about the sequence generated by the adversary. Here, rather than directly minimizing the learner's number of mistakes, we compare the learner's performance to that of the best concept within the concept class, a standard performance measure known as regret. Additionally, we note that when the learner's predictions are randomized, we focus on the expected values of the mentioned objectives.

In this paper, our main contribution is constructively answering the following questions in the list transductive online learning framework:

- What are the possible rates of the minimax number of mistakes in the realizable setting as a function of the concept class $\mathcal{C}$ and the size of the initial sequence selected by the adversary T?

- What is the necessary and sufficient combinatorial condition to make learnability possible in the agnostic regime?

Before this study, the questions outlined above remained entirely open, particularly in scenarios where the number of labels is unbounded. In fact, we show that there exists a concept class that is not learnable in the list online learning framework of (Moran et al., 2023), which does not assume prior knowledge of the sequence of instances, but it becomes learnable in the list transductive online learning framework with only a few mistakes Proposition C.5. Notably, the special case of list size of one, which is equivalent to the multiclass transductive online learning framework, was recently studied in (Hanneke et al., 2024b). In particular, our

results answer an open question posed in their paper, regarding the generalization of their results to the list setting. Moreover, we also demonstrate the existence of a concept class that is not learnable in the multiclass transductive online learning framework; however, again, it is learnable in the list transductive online learning framework with only a few mistakes by using a list size of two Proposition C.3. To complete our findings, we illustrate an example of another concept class, which is easily list PAC learnable, but it is not list transductive online learnable, showing that the finiteness of the dimension from the work of (Charikar & Pabbaraju, 2023) is not sufficient for list transductive online learnability Proposition C.4. Finally, we note that in the way of proving the agnostic result, we also resolve another open question asked by (Moran et al., 2023), regarding the extension of their agnostic result to the unbounded label spaces.

### 1.1. Related Work

**Online Learning.** Online learning has been a subject of study for more than half a century, gaining significant attention within the computer science community since the seminal work of (Littlestone, 1988). This foundational contribution introduced the adversarial online learning framework for binary classification setting, where during each round, an adversary selects an instance; afterward, the learner is required to predict a binary label for that instance; following this, the adversary reveals the correct label. The celebrated work of (Littlestone, 1988) also provided a combinatorial characterization of learnability for the mentioned problem in the realizable setting based on the Littlestone dimension. Later, (Ben-David et al., 2009) extended Littlestone's result to the agnostic setting, showing that the Littlestone dimension continues to characterize the learnability. Since then, online learning has been explored in various frameworks, including multiclass setting (Daniely et al., 2012; Hanneke et al., 2023a), and list setting (Moran et al., 2023). Given its fundamental nature, online learning has found numerous practical applications.

**Transductive Online Learning.** In contrast to the above definition of online learning, an alternative setting involves a scenario where the sequence of instances is predetermined by an adversary before the start of the game. This setting can eliminate the uncertainty associated with instances, yet retains the uncertainty about the labels. The study of this setting was first initiated by (Ben-David et al., 1997), with the goal of exploring how uncertainty in labeling alone influences the optimal number of mistakes. Furthermore, given that the online learnable classes are quite limiting, it is natural to extend the learnable classes whenever we have additional assumptions. Notably, the recent work of (Hanneke et al., 2023b) referred to this setting by the term "Transductive Online Learning" due to its relation to transductive PAC learning. Given the set of instances before the

start of the game, other related lines of research include the self-directed online learning framework, in which the learning algorithm is permitted to choose the next instance for prediction from the remaining set of instances in each round. Additionally, there is the best order framework, where the learner, rather than an adversary, determines the order of instances at the outset of the game. See (Goldman & Sloan, 1994; Ben-David et al., 1995; Ben-David & Eiron, 1998; Devulapalli & Hanneke, 2024) for more details.

## 1.2. Overview of the Main Results

In the subsequent subsection, we provide an overview of the primary results in our paper along with a summary of the proof techniques.

### 1.2.1. LIST TRANSDUCTIVE ONLINE LEARNING FRAMEWORK

We consider a sequential game between the learner and an adversary over a total of $T \in \mathbb{N}$ rounds. Initially, an adversary chooses a sequence of T instances from a non-empty instance space $\mathcal{X}$, namely $(x_1, x_2, \ldots, x_T)$ and reveals it to the learner. Moreover, at each round $t \in [T]$, the adversary selects a label $y_t$ from a non-empty label space $\mathcal{Y}$, which can possibly be uncountable; then, the learner predicts a list of size $L \in \mathbb{N}$ of labels, which can be randomized; subsequently, the learner, observes the true label $y_t$. Before going forward, following the well-established frameworks in learning theory, we consider a concept class $\mathcal{C}$ as a set of mappings from $\mathcal{X}$ to $\mathcal{Y}$ that is known to the learner before starting the game. See subsubsection 2.2.1 and subsubsection 2.2.2 for more details.

### 1.2.2. REALIZABLE SETTING

In the realizable setting, we assume that the sequence $(x_1, y_1), \ldots, (x_T, y_T)$, generated by the adversary, is consistent with at least one mapping in the concept class $\mathcal{C}$. Moreover, in this setting, we focus on the standard notion of the number of mistakes made by the learner over T rounds. In particular, we aim to establish upper and lower bounds on the minimax number of mistakes, as a function of $\mathcal{Q}$ as an instance of the list transductive online learning framework and T as a total number of rounds, denoted by $M^\star(\mathcal{Q}, T)$. See subsubsection 2.2.4 for more details.

Our primary result in this part demonstrates that if the learner plays for $T \in \mathbb{N}$ rounds, its minimax number of mistakes can only be of the orders $\Theta(T)$, $\Theta(\log T)$, or $\Theta(1)$. Furthermore, this trichotomy is fully characterized by the finiteness of the Level-constrained $(L + 1)$-Littlestone dimension and the Level-constrained $(L + 1)$-Branching dimension, where the list size is $L \in \mathbb{N}$.

To define the Level-constrained $(L + 1)$-Littlestone dimen-

sion, we first need to define the Level-constrained $(L + 1)$-Littlestone tree. A Level-constrained $(L + 1)$-Littlestone tree is a $(L + 1)$-Littlestone tree with the additional requirement that for a given level the same instance has to label all nodes. Then, the $(L + 1)$-Level-constrained Littlestone dimension is defined as $\sup_{d \in \mathbb{N}}$ such that there exists a shattered Level-constrained $(L + 1)$-Littlestone tree of depth $d$. To define the Level-constrained $(L + 1)$-Branching dimension, we first need to define the Level-constrained $(L + 1)$-Branching tree. A Level-constrained $(L + 1)$-Branching tree is a Level-constrained $(L + 1)$-Littlestone tree without the restriction that the labels on the two outgoing edges are distinct. Then, the Level-constrained $(L + 1)$-Branching dimension is defined as $\sup_{d \in \mathbb{N}}$ such that there exists a shattered Level-constrained $(L + 1)$-Branching tree such that every root-to-leaf path contains at least $d$ nodes with all distinct labels. See subsection 2.3 for more details. Formally, we have the following theorem.

**Theorem 1.1.** *Let* $\mathcal{Q} = \big( \mathcal{X}, L, \mathcal{Y}, \mathcal{C} \big)$ *be an instance of the list transductive online learning framework. Then, we have:*

$$
M^\star(\mathcal{Q}, T) \in \begin{cases} \Theta(1), & B(\mathcal{Q}) < \infty \\ \Theta(\log T), & D(\mathcal{Q}) < \infty \text{ and } B(\mathcal{Q}) = \infty \\ \Theta(T), & D(\mathcal{Q}) = \infty \end{cases} ,
$$

*where* $B(\mathcal{Q})$ *is the Level-constrained* L*-Branching dimension of* $\mathcal{Q}$, *and* $D(\mathcal{Q})$ *is the Level-constrained* L*-Littlestone dimension of* $\mathcal{Q}$, *both of them defined in subsection 2.3.*

The proof of the above theorem comprises up several components. First, to establish the upper bound in the constant case, basically, we generalize the notion of rank introduced by (Ben-David et al., 1997) to the list setting. Then, we use the adaptation of Littlestone's Standard Optimal Algorithm (SOA) to get the final result. Next, we derive two lower bounds by using ideas from (Hanneke et al., 2024b). For the upper bound in the $\log T$ case, the first idea that one may think of is to employ the Halving algorithm combined with the list Sauer-Shelah-Perles (SSP) Lemma from (Charikar & Pabbaraju, 2023; Hanneke et al., 2024a). However, as discussed by (Hanneke et al., 2024b), this approach is inapplicable when the label space is unbounded. Importantly, even when focusing on the finite label space setting, this approach is not applicable at all. To see this, notice that the number of functions based on the list SSP Lemma from (Charikar & Pabbaraju, 2023; Hanneke et al., 2024a), can be of order $\mathcal{O}(L^T T^d k^{L d})$, where $k$ denotes the size of the label space, $L$ is the list size, and $d$ is the associated dimension. Consequently, as the reduction in the version space per mistake is only by a factor of $1 - L / k$, this approach leads to a bound that is linear in T. To overcome this obstacle, we generalize the technique of (Hanneke et al., 2024b). This technique enables us to establish the desired upper bound of $\log T$ on the minimax number of mistakes. In summary,

we define a notion of shattering for a sequence of instances from $\mathcal{X}$. Based on the finiteness of $\mathrm{D}(\mathcal{Q})$, we can bound the total number of sub-sequences of the initial sequence played by the adversary that are shattered. A key observation is that if we do not have any shattered sub-sequence, we will make no more mistakes. Our algorithm can guarantee a decrease in the number of shattered sub-sequences after making a mistake, so the total number of mistakes is bounded. See section 3 and Appendix A for more details.

Before proceeding, we emphasize that in contrast to the multiclass setting, even the finite label space regime for list learning cannot be handled by the simple approach of Halving via the SSP Lemma. Moreover, there are different ways of selecting L labels. We found that decreasingly sorting the label set $\mathcal{Y}$ based on appropriate measures and selecting the first L labels works in both of our upper bounds. Also, we have the following inequalities: $\mathrm{DS}(\mathcal{Q}) \leq \mathrm{D}(\mathcal{Q}) \leq \mathrm{B}(\mathcal{Q}) \leq \mathrm{LD}(\mathcal{Q})$; see Appendix C.

### 1.2.3. AGNOSTIC SETTING

In the agnostic setting, we make no assumptions about the sequence $(x_1, y_1), (x_2, y_2), \ldots, (x_{\mathrm{T}}, y_{\mathrm{T}})$, generated by the adversary. Moreover, in this setting, our focus shifts to the standard notion of expected regret, which compares the expected number of mistakes made by the learner to those made by the best concept in the concept class $\mathcal{C}$ over the sequence. In particular, we say that an instance of the list transductive online learning framework $\mathcal{Q}$ is agnostic learnable in the list transductive online learning framework, if the minimax expected regret, as a function of $\mathcal{Q}$ and T as a total number of rounds, denoted by $\mathbf{R}^\star(\mathcal{Q}, \mathrm{T})$, is sub-linear in T. See subsubsection 2.2.5 for more details.

Our primary result in this part demonstrates that this criterion for learnability is fully characterized by the finiteness of the Level-constrained $(\mathrm{L}+1)$-Littlestone dimension, if the list size is $\mathrm{L} \in \mathbb{N}$. Formally, we have the following theorem.

**Theorem 1.2.** *Let $\mathcal{Q} = (\mathcal{X}, \mathrm{L}, \mathcal{Y}, \mathcal{C})$ be an instance of the list transductive online learning framework. Then, $\mathcal{Q}$ is agnostic learnable in the list transductive online learning framework if and only if it has finite Level-constrained $(\mathrm{L}+1)$-Littlestone dimension.*

The proof of the above theorem involves a few pieces. First, we establish the lower bound for randomized learners by leveraging ideas from (Moran et al., 2023). To prove the upper bound, we face the challenge of dealing with an unbounded label space, which renders the standard techniques from (Ben-David et al., 2009; Daniely et al., 2012) inapplicable. To overcome this obstacle, we employ the dynamic expert technique from the recent work of (Hanneke et al., 2023a). To do so, we use our technique from the realizable

setting for proving the upper bound in the $\log(\mathrm{T})$ case. Furthermore, we combined dynamic experts with the celebrated exponential weights algorithm to get the final result. Our finding indicates that the technique introduced in (Hanneke et al., 2023a) effectively addresses infinite label spaces even in the list setting, thereby resolving the question raised by (Moran et al., 2023) regarding the extension of their agnostic result to the unbounded label spaces. See section 4 and Appendix B for more details.

### 1.3. Organization

The rest of the paper is organized as follows. In section 2, we formally set the notation and definitions. Subsequently, in section 3, we present our results for the realizable setting. Then, in section 4, we extend our results to the agnostic setting. Afterward, in Appendix C, we provide a few examples showing separations between different related combinatorial complexity dimensions. Eventually, in section 5, we conclude our manuscript and present some future directions.

## 2. Notations, Definitions, and Preliminaries

In this section, we set our basic notations in subsection 2.1. Then, we present the List Transductive Online Learning framework in subsection 2.2. Finally, we define main combinatorial complexity measures in this paper in subsection 2.3.

### 2.1. Notations

In this subsection, we present the basic notations that we use throughout our paper. Let $\mathbb{N}$ and $\mathbb{R}$ stand for the set of natural and real numbers, accordingly. We denote by $\bar{\mathbb{N}}$ the extended natural number system defined as $\bar{\mathbb{N}} := \mathbb{N} \cup \{-\infty, +\infty\}$. Also, for a given $n \in \mathbb{N}$, we use $[n]$ to denote $\{1, 2, \ldots, n\}$. Next, let $n \in \mathbb{N}$, for any sequence of size $n$ or $n$-tuple $x$, and any $i \in \mathbb{N}$ such that $1 \leq i \leq n$, let us use $x_i$ to denote the $i$-th element in $x$. To increase the readability of our manuscript, we use "," to separate indices of elements when we have more than one index; for instance, let $x$ be a sequence of size 5 of 2-tuples, we denote by $x_{5,1}$ the first element of the 5-th element of $x$. We denote by $A \times B$ the Cartesian product of two arbitrarily sets $A$ and $B$. In addition, for any set $A$ and any $n \in \mathbb{N}$, we let $A^n$ indicate $n$ times the Cartesian product of $A$ with itself. Note that, for any set $A$, we define $A^0 := \{\emptyset\}$. Also, given a set $A$, we denote by $A^\star$ the set of all finite sequences of the members of $A$; more formally, $A^\star := \bigcup_{T=0}^{\infty} A^T$. Then, for the arbitrary sets $X$ and $Y$, we use $Y^X$ to denote the space of all functions from $X$ to $Y$. Finally, we use $\mathcal{O}(.), o(.), \Omega(.), \omega(.),$ and $\Theta(.)$ as standard notations of them in the theoretical computer science. We also use $\widetilde{\mathcal{O}}(.), \widetilde{\Omega}(.), \widetilde{\Theta}(.)$ to exclude logarithmic factors as well as constant coefficients.

## 2.2. List Transductive Online Learning Framework

In this subsection, we present the List Transductive Online Learning framework. To do so, first, we give the problem setup in subsection 2.2.1. Then, we formulate the problem in subsection 2.2.2. Afterward, we define list transductive online learning algorithms in subsection 2.2.3. Finally, we give the associated definitions with the realizable and agnostic settings in subsection 2.2.4 and subsection 2.2.5, accordingly.

### 2.2.1. PROBLEM SETUP

Fix a non-empty set $\mathcal{X}$ as the instance space. Let $L \in \mathbb{N}$ be the size of the list. Also, fix a non-empty set $\mathcal{Y}$ equipped with a $\sigma$-algebra such that every subset of $\mathcal{Y}$ with cardinality L is measurable as the label space. We note briefly that, since our focus is on deterministic algorithms in the realizable setting, no measurability assumptions on $\mathcal{Y}$ are required. Following the well-established frameworks in learning theory, let $\mathcal{C} \subseteq \mathcal{Y}^{\mathcal{X}}$ be a concept class. In particular, a 4-tuple $\mathcal{Q} = (\mathcal{X}, L, \mathcal{Y}, \mathcal{C})$ presents an instance of the list transductive online learning framework.

### 2.2.2. LIST TRANSDUCTIVE ONLINE LEARNING GAME

Let $T \in \mathbb{N}$. The problem of list transductive online learning is formulated as a T-rounded sequential game between the learner/player and an adversary/opponent. Initially, an adversary chooses a sequence of T instances $X \in \mathcal{X}^T$ and reveals it to the learner. Moreover, at each round $t \in [T]$:

- The adversary chooses a label $y_t$ from $\mathcal{Y}$.

- The learner predicts a list of size L of labels.

- The adversary reveals the true label $y_t$.

### 2.2.3. LIST TRANSDUCTIVE ONLINE LEARNING RULES

We consider two different types of list transductive online learning rules/algorithms, namely deterministic rules and randomized rules. As a result, we have the following definitions.

**Definition 2.1** (Deterministic List Transductive Online Learning Rule). Let $\mathcal{D} = \{(x^\star, y^\star) \mid x^\star \in \mathcal{X}^\star, \; y^\star \in \mathcal{Y}^\star, \; |y^\star| < |x^\star|\}$. In addition, let $\mathcal{Y}_L = \{\mathcal{A} \mid \mathcal{A} \subseteq \mathcal{Y}, \; |\mathcal{A}| = L\}$. A deterministic list transductive online learning rule is a mapping $\mathbf{A} : \mathcal{D} \to \mathcal{Y}_L$.

In words, it is a mapping that maps each finite sequence of instances and a finite sequence of labels with a size smaller than the size of the sequence of instances to a set of size L of labels.

**Definition 2.2** (Randomized List Transductive Online Learning Rule). Let $\mathcal{Y}_L = \{\mathcal{A} \mid \mathcal{A} \subseteq \mathcal{Y}, \; |\mathcal{A}| = L\}$. In addition, let $\mathcal{D} = \{(x^\star, a^\star, y^\star) \mid x^\star \in \mathcal{X}^\star, \; a^\star \in \mathcal{A}^\star, \; y^\star \in$

$\mathcal{Y}^\star, \; |a^\star| = |y^\star| < |x^\star|\}$. A randomized list transductive online learning rule is a mapping $\mathbf{A} : \mathcal{D} \to \Pi(\mathcal{Y}_L)$.

In words, it is a mapping that maps each finite sequence of instances denoted by $x'$, a finite sequence of set of labels of size L denoted by $a'$, and a finite sequence of labels denoted by $y'$ such that $|a^\star| = |y^\star| < |x^\star|$ to a probability measure on $\mathcal{Y}_L$.

### 2.2.4. REALIZABLE SETTING

Here, we begin by defining a realizable sequence. Then, to evaluate the performance of any deterministic algorithm, we define the well-known notion of the number of mistakes adapted to our framework. Finally, we define the optimal mistake bound, building on the previous definitions.

**Definition 2.3** (Realizable Sequence). Fix $T \in \mathbb{N}$. We say that a finite sequence of size T of instance-label pairs $\big((x_1, y_1), (x_2, y_2), \ldots, (x_T, y_T)\big) \in (\mathcal{X} \times \mathcal{Y})^T$ is realizable by a concept class $\mathcal{C}$ if there exists a concept $c \in \mathcal{C}$ such that for every $i \in [T]$, we have $c(x_i) = y_i$.

**Definition 2.4** (Number of Mistakes). Let $\mathbf{A}$ be a deterministic list transductive online learning rule. Fix $T \in \mathbb{N}$. Let $S$ be a finite sequence of size T of instance-label pairs $S = \big((x_1, y_1), (x_2, y_2), \ldots, (x_T, y_T)\big) \in (\mathcal{X} \times \mathcal{Y})^T$. We define the number of mistakes made by $\mathbf{A}$ with respect to the sequence $S$, denoted by $\mathbf{M}(\mathbf{A}; S)$, as follows:

$$\mathbf{M}(\mathbf{A}; S) := \sum_{t=1}^{T} \mathbb{1}\Big\{ y_t \notin \mathbf{A}\big((\mathcal{S}', (y_1, y_2, \ldots, y_{t-1}))\big) \Big\},$$

where $\mathcal{S}'$ is defined as follows: $\mathcal{S}' := (x_1, x_2, \ldots, x_T)$.

**Definition 2.5** (Optimal Mistake Bound). Let $\mathcal{Q} = (\mathcal{X}, L, \mathcal{Y}, \mathcal{C})$ be an instance of the list transductive online learning framework. The optimal mistake bound of $\mathcal{Q}$ as a function of the time horizon T, denoted by $\mathbf{M}^\star(\mathcal{Q}, T)$, is defined as follows:

$$\mathbf{M}^\star(\mathcal{Q}, T) := \inf_{\mathbf{A} \in \mathcal{A}} \sup_{\substack{S \in (\mathcal{X} \times \mathcal{Y})^T \text{ which is} \\ \text{realizable by } \mathcal{C}}} \mathbf{M}(\mathbf{A}; S),$$

where $\mathcal{A}$ is defined as the set of all deterministic list transductive online learning rules.

Before proceeding, it is important to note that while our definitions are based on an oblivious adversary, it is straightforward to see that they are equivalent to the case with an adaptive adversary.

### 2.2.5. AGNOSTIC SETTING

In this subsubsection, we begin by defining the well-known game theoretic notion of regret. Then, we present the definition of agnostic learnability.

**Definition 2.6** (Expected Regret). Let $\mathbf{A}$ be a randomized list transductive online learning rule. Fix $T \in \mathbb{N}$. Let $S$ be a finite sequence of size $T$ of instance-label pairs $S = \big((x_1, y_1), (x_2, y_2), \ldots, (x_T, y_T)\big) \in (\mathcal{X} \times \mathcal{Y})^T$. We define the expected regret of $\mathbf{A}$ with respect to the sequence $S$ against the competitor concept class $\mathcal{C}$, denoted by $\mathbf{R}(\mathbf{A}; S; \mathcal{C})$, as follows:

$$\mathbf{R}(\mathbf{A}; S; \mathcal{C}) := \mathbb{E}_{\mathbf{A}'}\Big[\mathbf{M}(\mathbf{A}'; S)\Big] - \inf_{c \in \mathcal{C}} \sum_{t=1}^{T} \mathbb{1}\{y_t \neq c(x_t)\},$$

where we view a single run of the randomized list transductive online learning rule $\mathbf{A}$ as running a deterministic list transductive online learning rule $\mathbf{A}'$. Moreover, we take the expectation over a random sequence of outputs of $\mathbf{A}$.

**Definition 2.7** (Optimal Expected Regret). Let $\mathcal{Q} = \big(\mathcal{X}, L, \mathcal{Y}, \mathcal{C}\big)$ be an instance of the list transductive online learning framework. The optimal expected regret bound of $\mathcal{Q}$ as a function of the time horizon $T$, denoted by $\mathbf{R}^\star(\mathcal{Q}, T)$, is defined as follows:

$$\mathbf{R}^\star(\mathcal{Q}, T) := \inf_{\mathbf{A} \in \mathcal{A}} \sup_{S \in (\mathcal{X} \times \mathcal{Y})^T} \mathbf{R}(\mathbf{A}; S; \mathcal{C}),$$

where $\mathcal{A}$ is defined as the set of all randomized list transductive online learning rules.

**Definition 2.8** (Agnostic Learnability). We say that an instance of the list transductive online learning framework $\mathcal{Q} = \big(\mathcal{X}, L, \mathcal{Y}, \mathcal{C}\big)$ is agnostic learnable in the list transductive online learning framework, if $\mathbf{R}^\star(\mathcal{Q}, T)$ as a function of the time horizon $T$ is sub-linear in $T$.

Before proceeding, it is important to note that while our definitions are based on an oblivious adversary, it is straightforward to see that they are equivalent to the case with an adaptive adversary. See Lemma 4.1 in (Cesa-Bianchi & Lugosi, 2006).

We emphasize that one may also consider a similar notion of learnability defined using $o(T)$ in the realizable setting.

## 2.3. Combinatorial Complexity Parameters

In this subsection, we first set our notations for trees. Then, we proceed with the definitions of the main combinatorial complexity parameters in our paper based on previous definitions.

**Definition 2.9** (Perfect Rooted L-ary Trees). Let $L \in \mathbb{N}$. A perfect rooted L-ary tree $\mathcal{T}$ is a rooted tree, each of whose internal nodes has exactly $L$ children and all leaves have the same depth.

**Definition 2.10** (L-ary $(\mathcal{X}, \mathcal{Y})$-valued Trees). Let $L \in \mathbb{N}$. Also, let $\mathcal{X}, \mathcal{Y}$ be any non-empty sets. A L-ary $(\mathcal{X}, \mathcal{Y})$-valued tree $\mathcal{T}$ is a perfect rooted L-ary tree, each of whose nodes are labeled by an element of $\mathcal{X}$, and each of whose edges are labeled by an element of $\mathcal{Y}$. Moreover, for any L-ary $(\mathcal{X}, \mathcal{Y})$-valued tree, a root to leaf path of length $\ell \in \mathbb{N}$ can be identified by a sequence of pairs $\mathfrak{s} \in (\mathcal{X} \times \mathcal{Y})^\ell$.

### 2.3.1. $(L+1)$-LITTLESTONE DIMENSION

**Definition 2.11** ($(L+1)$-Littlestone Tree). Let $\mathcal{Q} = \big(\mathcal{X}, L, \mathcal{Y}, \mathcal{C}\big)$ be an instance of the list transductive online learning framework. An $(L+1)$-ary $(\mathcal{X}, \mathcal{Y})$-valued tree $\mathcal{T}$ is called $(L+1)$-Littlestone tree for $\mathcal{Q}$.

**Definition 2.12** (Shattered $(L+1)$-Littlestone Tree). Let $\mathcal{Q} = \big(\mathcal{X}, L, \mathcal{Y}, \mathcal{C}\big)$ be an instance of the list transductive online learning framework. We say that a $(L+1)$-Littlestone tree for $\mathcal{T}$ for $\mathcal{Q}$ is shattered by $\mathcal{C}$, if for every finite root to leaf path in $\mathcal{T}$, identified by $\mathfrak{s} \in (\mathcal{X} \times \mathcal{Y})^\ell$ for some $\ell \in \mathbb{N}$, there exists a concept $c \in \mathcal{C}$ such that for every $i \in \mathbb{N}, i \leq \ell$, we have: $\mathfrak{s}_{i,2} = c(\mathfrak{s}_{i,1})$.

**Definition 2.13** ($(L+1)$-Littlestone Dimension). Let $\mathcal{Q} = \big(\mathcal{X}, L, \mathcal{Y}, \mathcal{C}\big)$ be an instance of the list transductive online learning framework. The $(L+1)$-Littlestone dimension of $\mathcal{Q}$, denoted by $L(\mathcal{Q})$, is defined as a $\sup_{d \in \bar{\mathbb{N}}}$ such that there exists a $(L+1)$-Littlestone tree $\mathcal{T}$ of depth $d$ for $\mathcal{Q}$ that all children of every node have distinct labels which is shattered by $\mathcal{C}$. Also, if $\mathcal{C} = \{\emptyset\}$, we have: $L(\mathcal{Q}) = 0$.

### 2.3.2. LEVEL-CONSTRAINED $(L+1)$-LITTLESTONE DIMENSION

**Definition 2.14** (Level-constrained $(L+1)$-Littlestone Dimension). Let $\mathcal{Q} = \big(\mathcal{X}, L, \mathcal{Y}, \mathcal{C}\big)$ be an instance of the list transductive online learning framework. The Level-constrained $(L+1)$-Littlestone dimension of $\mathcal{Q}$, denoted by $D(\mathcal{Q}) \in \bar{\mathbb{N}}$, is defined as a $\sup_{d \in \mathbb{N}}$ such that there exists a $(L+1)$-Littlestone tree $\mathcal{T}$ of depth $d$ for $\mathcal{Q}$ that all children of every node are labeled by distinct elements of $\mathcal{Y}$ and all nodes at the same level are labeled by the same element of $\mathcal{X}$ which is shattered by $\mathcal{C}$. Also, if $\mathcal{C} = \{\emptyset\}$, we have: $D(\mathcal{Q}) = 0$.

### 2.3.3. LEVEL-CONSTRAINED $(L+1)$-BRANCHING DIMENSION

**Definition 2.15** (Level-constrained $(L+1)$-Branching Dimension). Let $\mathcal{Q} = \big(\mathcal{X}, L, \mathcal{Y}, \mathcal{C}\big)$ be an instance of the list transductive online learning framework. The Level-constrained $(L+1)$-Branching dimension of $\mathcal{Q}$ denoted by $B(\mathcal{Q}) \in \bar{\mathbb{N}}$, is defined as a $\sup_{d \in \mathbb{N}}$ such that there exists a $(L+1)$-Littlestone tree $\mathcal{T}$ for $\mathcal{Q}$ that all nodes at the same level are labeled by the same element of $\mathcal{X}$ and every root to leaf path contains at least $d$ nodes labeled by distinct elements of $\mathcal{Y}$ which is shattered by $\mathcal{C}$. Also, if $\mathcal{C} = \{\emptyset\}$, we have: $B(\mathcal{Q}) = 0$.

# 3. Realizable Setting

First, we restate the main theorem in the realizable setting here for the sake of simplicity.

**Theorem 3.1.** *Let* $\mathcal{Q} = (\mathcal{X}, \mathrm{L}, \mathcal{Y}, \mathcal{C})$ *be an instance of the list transductive online learning framework. Then, we have:*

$$\mathbf{M}^{\star}(\mathcal{Q}, T) \in \begin{cases} \Theta(1), & \mathrm{B}(\mathcal{Q}) < \infty \\ \Theta(\log T), & \mathrm{D}(\mathcal{Q}) < \infty \ and \ \mathrm{B}(\mathcal{Q}) = \infty \ , \\ \Theta(T), & \mathrm{D}(\mathcal{Q}) = \infty \end{cases}$$

*Proof.* The theorem can be proved by combining two lower bounds of Lemma 3.2 and Lemma 3.5 as well as two upper bounds from Lemma A.1 and Lemma A.3. This finishes the proof. □

We provide the proofs for the lower bounds in the main text, while deferring the more technical proofs for the upper bounds to Appendix A.

## 3.1. Lower Bound T

We start by proving the lower bound for the linear case. Furthermore, the proof of the following Lemma uses the main idea in the lower bound proof of the work of (Littlestone, 1988).

**Lemma 3.2.** *Let* $\mathcal{Q} = (\mathcal{X}, \mathrm{L}, \mathcal{Y}, \mathcal{C})$ *be an instance of the list transductive online learning framework. Assume that* $\mathrm{D}(\mathcal{Q}) = \infty$. *Then, we have* $\mathbf{M}^{\star}(\mathcal{Q}, T) \in \Omega(T)$.

*Proof.* Fix $T \in \mathbb{N}$. Let $\mathcal{T}$ be a $(\mathrm{L} + 1)$-Littlestone tree witnessing $\mathrm{D}(\mathcal{Q}) = \mathrm{T}$. Note that such a tree should exist as $\mathrm{D}(\mathcal{Q}) = \infty$. Based on Definition 2.14, this tree should have depth T as well. Also, at each level of $\mathcal{T}$, all nodes are labeled by the same instance from $\mathcal{X}$. In addition, children of all nodes are labeled by distinct labels from $\mathcal{Y}$. Let $\mathbf{A}$ be any deterministic list transductive online learning rule. Now, we will build an adversarial strategy against $\mathbf{A}$ using $\mathcal{T}$. To do so, we first present T instances at T levels of $\mathcal{T}$ in order to the learner before starting the game. Furthermore, we will continue to make this strategy based on a special root-to-leaf path in $\mathcal{T}$, which depends on $\mathbf{A}$. Receive the first set of labels of size L predicted by the learner. As we have $\mathrm{L} + 1$ edges for the root node of $\mathcal{T}$ labeled with distinct labels from $\mathcal{Y}$, there exists at least one of them, which is labeled by a label that is not in the received set. We output that label. Moreover, we will continue this construction of the adversarial strategy based on $\mathcal{T}$ and $\mathbf{A}$. Indeed, we can force T number of mistakes to the $\mathbf{A}$. In addition, as $\mathcal{T}$ is shattered by $\mathcal{C}$, there exists a concept in $\mathcal{C}$, which is consistent with the root-to-leaf path that we used. So, we are in the realizable setting. As a result, we have: $\sup_{\substack{S \in (\mathcal{X} \times \mathcal{Y})^{\mathrm{T}} \text{ which is} \\ \text{realizable by } \mathcal{C}}} \mathbf{M}(\mathbf{A}; S) \geq \mathrm{T}$. Since we are able to

construct an adversarial strategy for an arbitrary deterministic list transductive online learning rule $\mathbf{A}$, we conclude $\mathbf{M}^{\star}(\mathcal{Q}, \mathrm{T}) \geq \mathrm{T}$. Finally, note that this argument works for any $\mathrm{T} \in \mathbb{N}$. This finishes the proof. □

Fix $\mathrm{T} \in \mathbb{N}$. Indeed, for a given $\mathcal{Q} = (\mathcal{X}, \mathrm{L}, \mathcal{Y}, \mathcal{C})$, if $\mathrm{D}(\mathcal{Q}) = \mathrm{d}$, using the above proof, we can show $\min\{\mathrm{T}, \mathrm{d}\}$ lower bound.

## 3.2. Lower Bound $\frac{\log(\mathrm{L}\,\mathbf{T}+1)}{\log(\mathrm{L}+1)}$

We continue by proving the lower bound for the Logarithmic case. Furthermore, the proof requires the following combinatorial Lemma. Additionally, we establish a similar lower bound as the one we just proved before presenting the main result.

**Lemma 3.3.** *Let* $\mathcal{Q} = (\mathcal{X}, \mathrm{L}, \mathcal{Y}, \mathcal{C})$ *be an instance of the list transductive online learning framework. Assume that* $\mathrm{B}(\mathcal{Q}) = \mathrm{d}$ *for some* $\mathrm{d} \in \mathbb{N}$. *Then, there exists a* $(\mathrm{L} + 1)$-*Littlestone tree of maximum depth* $\dfrac{(\mathrm{L} + 1)^{\mathrm{d}} - 1}{\mathrm{L}}$ *witnessing* $\mathrm{B}(\mathcal{Q}) = \mathrm{d}$, *crucially, just using a subset of nodes in any tree witnessing* $\mathrm{B}(\mathcal{Q}) = \mathrm{d}$.

*Proof.* We prove this Lemma by induction. We start with the base case. In particular, assume $\mathrm{d} = 1$. Let $\mathcal{T}$ be a $(\mathrm{L}+1)$-Littlestone tree witnessing $\mathrm{B}(\mathcal{Q}) = 1$. Thus, in every root-leaf-path in $\mathcal{T}$, we should have at least one node whose outgoing edges are labeled by $(\mathrm{L} + 1)$ distinct values from $\mathcal{Y}$. Take one such node. Indeed, this node can itself witness $\mathrm{B}(\mathcal{Q}) = 1$. So, the depth is at most $1 \leq \dfrac{(\mathrm{L} + 1)^{1} - 1}{\mathrm{L}} = 1$. This finishes the proof of the base case. Now, assume the claim is true for $\mathrm{d} \in \mathbb{N}$. Subsequently, we prove the claim for $\mathrm{d} + 1$. More specifically, we show that if $\mathrm{B}(\mathcal{Q}) = \mathrm{d} + 1$, then there exists a $(\mathrm{L} + 1)$-Littlestone tree of maximum depth $\dfrac{(\mathrm{L} + 1)^{\mathrm{d}+1} - 1}{\mathrm{L}}$ witnessing $\mathrm{B}(\mathcal{Q}) = \mathrm{d} + 1$. Let $\mathcal{T}$ be a $(\mathrm{L} + 1)$-Littlestone tree witnessing $\mathrm{B}(\mathcal{Q}) = \mathrm{d} + 1$. Find the node with the minimum level in $\mathcal{T}$ whose outgoing edges are labeled by $(\mathrm{L} + 1)$ distinct values from $\mathcal{Y}$. Indeed, such a node should exist. Also, we have several of them, just take one. Now, denote $(\mathrm{L} + 1)$ sub-trees of that node by $\mathcal{T}_1, \mathcal{T}_2, \ldots, \mathcal{T}_{(\mathrm{L}+1)}$. Restrict our instance space $\mathcal{X}$ to instances on the levels of these sub-trees and call it $\mathcal{X}'$. Also, consider functions induced by the projection of $\mathcal{C}$ to $\mathcal{X}'$ and call it $\mathcal{C}'$. Let $\mathcal{Q}' := (\mathcal{X}', \mathrm{L}, \mathcal{Y}, \mathcal{C}')$. Based on our construction of $\mathcal{Q}'$, we have $\mathrm{B}(\mathcal{Q}') = \mathrm{d}$. Now, we apply the induction hypothesis and get $\mathcal{T}_1', \mathcal{T}_2', \ldots, \mathcal{T}_{(\mathrm{L}+1)}'$ such that each of them witness $\mathrm{B}(\mathcal{Q}') = \mathrm{d}$ and their depths are bounded above by $\dfrac{(\mathrm{L} + 1)^{\mathrm{d}} - 1}{\mathrm{L}}$. Subsequently, let us join our one node to these sub-trees. In particular, to keep the level -constraint property, the final depth is bound by

$(L+1) \times \frac{(L+1)^d - 1}{L} + 1 = \frac{(L+1)^{d+1} - 1}{L}$. As we have a node with $L + 1$ outgoing edges labeled by $L + 1$ distinct labels from $\mathcal{Y}$, the resulting tree can witness $B(\mathcal{Q}) = d + 1$. This completes the proof. $\square$

**Lemma 3.4.** *Let* $\mathcal{Q} = (\mathcal{X}, L, \mathcal{Y}, \mathcal{C})$ *be an instance of the list transductive online learning framework. Assume that* $B(\mathcal{Q}) = d$ *for some* $d \in \mathbb{N}$. *Then, we have* $M^\star(\mathcal{Q}, T) \geq d$ *for large enough T.*

*Proof.* Let $\mathcal{T}$ be a $(L + 1)$-Littlestone tree of depth $r \in \mathbb{N}$ witnessing $B(\mathcal{Q}) = d$. Based on Definition 2.15, at each level of $\mathcal{T}$, all nodes are labeled with the same instance from $\mathcal{X}$. Let $A$ be any deterministic list transductive online learning rule. Now, we will build an adversarial strategy against $A$ using $\mathcal{T}$. To do so, we first present $r$ instances at $r$ levels of $\mathcal{T}$ in order to the learner before starting the game. Furthermore, we will continue to make this strategy based on a special root-to-leaf path in $\mathcal{T}$, which depends on $A$. Receive the first set of labels of size $L$ predicted by the learner. We may have $L + 1$ edges for the root node of $\mathcal{T}$ labeled with distinct labels from $\mathcal{Y}$. If that is the case, then there exists at least one of them, which is labeled by a label that is not in the received set. We output that label. If that is not the case, just output any of them. Moreover, we will continue this construction of the adversarial strategy based on $\mathcal{T}$ and $A$. Indeed, we can force $d$ number of mistakes to the $A$. This is because every root-to-leaf path in $\mathcal{T}$ should contain at least $d$ nodes having $L+1$ outgoing edges labeled by distinct labels from $\mathcal{Y}$. In addition, as $\mathcal{T}$ is shattered by $\mathcal{C}$, there exists a concept in $\mathcal{C}$, which is consistent with the root-to-leaf path that we used. So, we are in the realizable setting. As a result, we have: $\sup_{\substack{S \in (\mathcal{X} \times \mathcal{Y})^T \text{ which is} \\ \text{realizable by } \mathcal{C}}} M(A; S) \geq d$.

Since we are able to construct an adversarial strategy for an arbitrary deterministic list transductive online learning rule $A$, we conclude $M^\star(\mathcal{Q}, T) \geq d$ for large enough $T = r$. This finishes the proof. $\square$

**Lemma 3.5.** *Let* $\mathcal{Q} = (\mathcal{X}, L, \mathcal{Y}, \mathcal{C})$ *be an instance of the list transductive online learning framework. Assume that* $B(\mathcal{Q}) = \infty$. *Then, we have* $M^\star(\mathcal{Q}, T) \in \Omega\left(\frac{\log(L\,T + 1)}{\log(L + 1)}\right) \in \Omega(\log T)$.

*Proof.* Fix $T \in \mathbb{N}$. Solve $\frac{(L + 1)^d - 1}{L} = T$ for $d$. So, the result is $\frac{\log(L\,T + 1)}{\log(L + 1)}$. Now, define $r := \left\lfloor \frac{\log(L\,T + 1)}{\log(L + 1)} \right\rfloor$. Let $\mathcal{T}$ be a $(L + 1)$-Littlestone tree witnessing $B(\mathcal{Q}) = r$. Note that such a tree should exist as $B(\mathcal{Q}) = \infty$. Now, apply Lemma 3.3 on $\mathcal{T}$. Thus, we can get a $(L + 1)$-Littlestone tree witnessing $B(\mathcal{Q}) = r$ whose depth is at most $T$. Subsequently, based on the proof of Lemma 3.4,

we have $M^\star(\mathcal{Q}, T) \geq r$. Finally, note that this argument works for any $T \in \mathbb{N}$. This concludes the proof. $\square$

## 4. Agnostic Setting

First, we restate the main theorem in the agnostic setting here for the sake of simplicity.

**Theorem 4.1.** *Let* $\mathcal{Q} = (\mathcal{X}, L, \mathcal{Y}, \mathcal{C})$ *be an instance of the list transductive online learning framework. Then,* $\mathcal{Q}$ *is agnostic learnable in the list transductive online learning framework if and only if* $D(\mathcal{Q}) < \infty$.

*Proof.* The theorem can be proved by combining a lower bound of Lemma B.1 and as well as an upper bound from Lemma B.2. $\square$

## 5. Conclusion, Discussion, and Future Directions

In this work, we investigated the problem of list transductive online learning with possibly arbitrary label space. In the realizable setting, we showed a trichotomy of possible minimax rates for the number of mistakes. In addition, we demonstrated a dichotomy of the minimax expected regret in the agnostic setting. To do so, we introduced two new combinatorial complexity parameters, the Level-constrained $(L + 1)$-Littlestone dimension and the Level-constrained $(L + 1)$-Branching dimension for some $L \in \mathbb{N}$.

Finally, we outline a potential future direction for this line of research. Similarly to our work, in all previous studies on list learnability, including (Charikar & Pabbaraju, 2023; Moran et al., 2023; Brukhim et al., 2024), additional factors related to the size of the list appear in the upper bounds. For instance, if the size of the list is $L \in \mathbb{N}$, in the work of (Charikar & Pabbaraju, 2023), a factor $L^6$ is present, or in the work of (Brukhim et al., 2024), a factor $L^4$ arises. A key open question is how to eliminate such factors from our $\log T$ upper bound in the realizable setting. Addressing this question could potentially lead to the elimination of list size factors in other related problems as well. In close relation to this question, one could explore the problem of list learning with possibly unbounded list size. For example, the label space can be the real numbers, while lists are intervals of size $c$ for some constant $c \in \mathbb{R}$.

## Impact Statement

This paper presents work whose goal is to advance the field of Statistical Learning Theory. There are many potential societal consequences of our work, none of which we feel must be specifically highlighted here.

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

---

Figure 1: List Transductive Standard Optimal Algorithm (LTSOA) is a variant of Standard Optimal Algorithm (SOA) originally proposed by (Littlestone, 1988). Further, see the definition of $\mathcal{V}_{x \to y}$ for some $\mathcal{V} \subseteq \mathcal{Y}^{\mathcal{X}}$, $x \in \mathcal{X}$, and $y \in \mathcal{Y}$ in the proof of Lemma A.1. In addition, see the definition of a sequence-dependent Level-constraint $(\mathrm{L} + 1)$-Branching dimension in the proof of Lemma A.1.

## A. Realizable Upper Bounds Proofs

### A.1. Upper Bound $\mathrm{B}(\mathcal{Q})$

Next, we move on to the proof of the constant upper bound. Furthermore, the proof of the following Lemma uses the main idea in the upper bound proof of the work of (Littlestone, 1988).

**Lemma A.1.** *Let $\mathcal{Q} = (\mathcal{X}, \mathrm{L}, \mathcal{Y}, \mathcal{C})$ be an instance of the list transductive online learning framework. Assume that $\mathrm{B}(\mathcal{Q}) = \mathrm{d}$. Then, we have $\mathbf{M}^\star(\mathcal{Q}, T) \in \mathcal{O}(\mathrm{d}) \in \mathcal{O}(1)$.*

*Proof.* We first define $\mathcal{V}_{x \to y}$ for some $\mathcal{V} \subseteq \mathcal{Y}^{\mathcal{X}}$, $x \in \mathcal{X}$, and $y \in \mathcal{Y}$. In particular, $\mathcal{V}_{x \to y} := \{c \mid c \in \mathcal{V}_{x \to y}, \ c(x) = y\}$. Next, for every $X \in \mathcal{X}^k$ for some $k \in \mathbb{N}$, we define $\mathrm{B}\left(\left(X, \mathrm{L}, \mathcal{Y}, \mathcal{C}\right)\right)$ by adding a new constraint to Definition 2.15. In particular, we require to use instances of $X$ in order as the label of the nodes of $(\mathrm{L} + 1)$-ary Littlestone tree.

Now, we proceed with the proof of the Lemma. Fix $\mathrm{T} \in \mathbb{N}$. We run LTSOA Algorithm 1 with input $\mathcal{Q}$, T, and $X \in \mathcal{X}^{\mathrm{T}}$ as the initial sequence of instances chosen by the adversary. First of all, notice that for any instance of the list transductive online learning framework $\mathcal{Q}'$, we have $\mathrm{B}(\mathcal{Q}') \geq 0$. Also, notice that $\mathrm{B}\left(\left((X_1, X_2, \ldots, X_{\mathrm{T}}), \mathrm{L}, \mathcal{Y}, \mathcal{C}\right)\right) \leq \mathrm{B}(\mathcal{Q})$. As a result, based on the following Claim, it is clear that $\sup_{S \in (\mathcal{X} \times \mathcal{Y})^{\mathrm{T}} \text{ which is realizable by } \mathcal{C}} \mathrm{M}(\mathrm{LTSOA}; S) \leq \mathrm{B}(\mathcal{Q}) = \mathrm{d}$. Therefore, $\mathbf{M}^\star(\mathcal{Q}, \mathrm{T}) \leq \mathrm{B}(\mathcal{Q}) = \mathrm{d}$. This completes the proof. Subsequently, we prove the following Claim.

*Claim* A.2. For every $t \in [\mathrm{T}]$, if $y_t \notin \mathcal{A}_t$, then $\mathrm{B}\left(\left((X_t, X_{t+1}, \ldots, X_{\mathrm{T}}), \mathrm{L}, \mathcal{Y}, \mathcal{V}^{t-1}\right)\right) > \mathrm{B}\left(\left((X_{t+1}, X_{t+2}, \ldots, X_{\mathrm{T}}), \mathrm{L}, \mathcal{Y}, \mathcal{V}^t\right)\right)$.

**Proof** We prove this claim by contradiction. For simplicity, denote $\mathrm{B}\left(\left((X_t, X_{t+1}, \ldots, X_{\mathrm{T}}), \mathrm{L}, \mathcal{Y}, \mathcal{V}^{t-1}\right)\right)$ by $A$. Also, denote $\mathrm{B}\left(\left((X_{t+1}, X_{t+2}, \ldots, X_{\mathrm{T}}), \mathrm{L}, \mathcal{Y}, \mathcal{V}^t\right)\right)$ by $B$. Assume that $A \geq B$. Indeed, the case that $A < B$ is not possible. So, assume that $A = B$. If that is the case, based on lines 1 and 2 in Algorithm 1, it means that there are at least $\mathrm{L} + 1$ labels such that the restriction of the current concept class $V^{t-1}$ to $x_t$ using those labels still leads to $A$ as a new dimension. Thus, this means that $A > A$ as we can construct a new tree. This is a clear contradiction. $\square$

---

**List Shattering Algorithm**

Input: A 4-tuple $\mathcal{Q} = (\mathcal{X}, \mathrm{L}, \mathcal{Y}, \mathcal{C})$, a total number of rounds $\mathrm{T} \in \mathbb{N}$, and a sequence of T instances $X \in \mathcal{X}^{\mathrm{T}}$.

Initialize $\mathcal{V}^0 = \mathcal{C}$ and $t = 1$.

While $(t \leq \mathrm{T})$:

1. Sort labels in $\mathcal{Y}$ in a non-increasing order according to the values $\mathrm{Sh}\left(\left((X_{t+1}, X_{t+2}, \ldots, X_{\mathrm{T}}), \mathrm{L}, \mathcal{Y}, \mathcal{V}^{t-1}_{x_t \to y}\right)\right)$ for $y \in \mathcal{Y}$.

2. Predict the list $\mathcal{A}_t$ which consists of the top L labels in the above order.

3. Receive a label $y_t \in \mathcal{Y}$.

4. Set $\mathcal{V}^t = V^{t-1}_{x_t \to y_t}$ and update $t = t + 1$.

---

Figure 2: See the definition of $\mathcal{V}_{x \to y}$ for some $\mathcal{V} \subseteq \mathcal{Y}^{\mathcal{X}}$, $x \in \mathcal{X}$, and $y \in \mathcal{Y}$ in the proof of Lemma A.3. In addition, see the definition of $\mathrm{Sh}(.)$ in the proof of Lemma A.3.

**A.2. Upper Bound** $\mathrm{L}\,\mathrm{D}(\mathcal{C}) \log\left(\frac{e\,\mathbf{T}}{\mathrm{D}(\mathcal{Q})}\right)$

Finally, we turn our attention to proving the $\log \mathrm{T}$ upper bound. Moreover, the proof of the following lemma represents the main contribution of this section, relying on the shattering technique from the recent work of (Hanneke et al., 2024b).

**Lemma A.3.** *Let $\mathcal{Q} = (\mathcal{X}, \mathrm{L}, \mathcal{Y}, \mathcal{C})$ be an instance of the list transductive online learning framework. Assume that* $\mathrm{D}(\mathcal{Q}) = \mathrm{d}$. *Then, we have* $\mathbf{M}^\star(\mathcal{Q}, T) \in \mathcal{O}\left(\mathrm{L}\,\mathrm{d}\log\left(\frac{e\,T}{\mathrm{d}}\right)\right) \in \log(T)$.

*Proof.* We first define $\mathcal{V}_{x \to y}$ for some $\mathcal{V} \subseteq \mathcal{Y}^{\mathcal{X}}$, $x \in \mathcal{X}$, and $y \in \mathcal{Y}$. In particular, $\mathcal{V}_{x \to y} := \{c \mid c \in \mathcal{V}_{x \to y},\ c(x) = y\}$. Next, for every $X \in \mathcal{X}^k$ for some $k \in \mathbb{N}$, we say that $X$ is full shattered by $\mathcal{C}$ if there exists a $(\mathrm{L}+1)$-ary Littlestone tree $\mathcal{T}$ of depth $k$ for $\mathcal{Q}$ such that all children of every node of $\mathcal{T}$ are labeled by distinct elements of $\mathcal{Y}$ and for every $i \in [k]$ all nodes at level $i$ of $\mathcal{T}$ are labeled by $X_i$ which is shattered by $\mathcal{C}$. Now, let $\mathcal{V} \subseteq \mathcal{Y}^{\mathcal{X}}$ and $X \in \mathcal{X}^k$ for some $k \in \mathbb{N}$. Then, we define $\mathrm{Sh}\left(\left((X_1, X_2, \ldots, X_k), \mathrm{L}, \mathcal{Y}, \mathcal{V}\right)\right)$ as the number of non-empty sub-sequences of $X$ that are full shattered by $\mathcal{V}$.

The first observation is if $X \in \mathcal{X}^\star$ is full shattered by $\mathcal{V} \subseteq \mathcal{C}$, then the size of $X$ should be smaller than $\mathrm{d}+1$. Next, clearly, if $X \in \mathcal{X}^\star$ does not have any non-empty sub-sequence that is full shattered by $\mathcal{V} \subseteq \mathcal{C}$, the projection of $\mathcal{V}$ on $X$ is unique.

Now, we proceed with the proof of the Lemma. Fix $\mathrm{T} \in \mathbb{N}$. We run List Shattering Algorithm 2 with input $\mathcal{Q}$, T, and $X \in \mathcal{X}^{\mathrm{T}}$ as the initial sequence of instances chosen by the adversary. Based on the previous paragraph, we know $\mathrm{Sh}\left(X, \mathrm{L}, \mathcal{Y}, \mathcal{V}\right) \leq \sum_{i=1}^{\mathrm{d}} \binom{\mathrm{T}}{i} \leq \left(\frac{e\,\mathrm{T}}{\mathrm{d}}\right)^{\mathrm{d}}$. Whenever we make a mistake, there are some cases to consider. (1) For each sub-sequence full shattered by all L of the version spaces related to the labels in the predicted list, and by the $y_t$ version space, there is another full shattered sequence being removed. In particular, the one that has the predicted point at its root and the other $\mathrm{L}+1$ trees as its sub-trees. (2) The remaining full shattered sub-sequences by the $y_t$ version space can be full shattered by at most $\mathrm{L}-1$ of the top L labels. Since those L labels all shattered at least as many sub-sequences as $y_t$, that should mean the maximum number of sub-sequences that can be full shattered by the $y_t$ version space is a $\left(1 - \frac{1}{\mathrm{L}}\right)$ fraction of the total full shattered sub-sequences. As a result, after at most $m > \frac{-\ln(A)}{\ln\left(1-\frac{1}{L}\right)}$ mistakes, where $A := \left(\frac{e\,\mathrm{T}}{\mathrm{d}}\right)^{\mathrm{d}}$, we have no full shattered sub-sequences. Therefore, we can get $\mathbf{M}^\star(\mathcal{Q}, \mathrm{T}) \in \mathcal{O}\left(\mathrm{L}\,\mathrm{d}\log\left(\frac{e\,\mathrm{T}}{\mathrm{d}}\right)\right) \in \log(\mathrm{T})$. $\qquad\square$

# B. Agnostic Proofs

## B.1. Lower Bound

We start by proving the lower bound for the linear case. Furthermore, the proof of the following Lemma uses the main idea in the lower bound proof of the work of (Moran et al., 2023).

**Lemma B.1.** *Let $\mathcal{Q} = (\mathcal{X}, \mathrm{L}, \mathcal{Y}, \mathcal{C})$ be an instance of the list transductive online learning framework. Assume that* $\mathrm{D}(\mathcal{Q}) = \infty$. *Then, $\mathcal{Q}$ is not agnostic learnable in the list transductive online learning framework.*

*Proof.* First of all, note that the realizable setting is a particular case of the agnostic setting. Fix $\mathrm{T} \in \mathbb{N}$. Let $\mathcal{T}$ be a $(\mathrm{L}+1)$-Littlestone tree witnessing $\mathrm{D}(\mathcal{Q}) = \mathrm{T}$. Note that such a tree should exist as $\mathrm{D}(\mathcal{Q}) = \infty$. Based on Definition 2.14, this tree should have depth $\mathrm{T}$ as well. Also, at each level of $\mathcal{T}$, all nodes are labeled by the same instance from $\mathcal{X}$. In addition, children of all nodes are labeled by distinct labels from $\mathcal{Y}$. Let $\mathbf{A}$ be any randomized list transductive online learning rule. Now, we will build an adversarial strategy against $\mathbf{A}$ using $\mathcal{T}$. To do so, we first present $\mathrm{T}$ instances at $\mathrm{T}$ levels of $\mathcal{T}$ in order to the learner before starting the game. Furthermore, we will continue to make this strategy based on a special root-to-leaf path in $\mathcal{T}$. Receive the first set of labels of size $\mathrm{L}$ predicted by the learner. As we have $\mathrm{L}+1$ edges for the root node of $\mathcal{T}$ labeled by distinct labels from $\mathcal{Y}$, there exists at least one of them, which is labeled by a label that is not in the received set. We output a random label from those labels according to the uniform distribution. Moreover, we will continue this construction of the adversarial strategy based on $\mathcal{T}$. Notice that, as $\mathcal{T}$ is shattered by $\mathcal{C}$, there exists a concept in $\mathcal{C}$, which is consistent with the root-to-leaf path that we used. So, we are in the realizable setting. In addition, notice that our choices of labels are completely independent of the learner's prediction. So, $\mathbf{A}$ makes a mistake at every point with probability of at least $\dfrac{1}{\mathrm{L}+1}$. Since we are able to construct an adversarial strategy for an arbitrary randomized list transductive online learning rule $\mathbf{A}$, we conclude any randomized rule has at least $\dfrac{\mathrm{T}}{\mathrm{L}+1}$ expected regret. It means that $\mathcal{Q}$ is not agnostic learnable in the list transductive online learning framework. This finishes the proof. $\quad\square$

## B.2. Upper Bound

Finally, we turn our attention to proving the $\sqrt{\mathrm{T}} \times \log(\mathrm{T})$ upper bound. Moreover, the proof of the following lemma represents the main contribution of this section, relying on the shattering technique from the realizable part, as well as the proof technique of (Hanneke et al., 2023a).

We note that Algorithm 2 can be made conservative, that is, only adding an running the line 5 if it is a mistake. It is not hard to see that we can get the same guarantee as in A.3. Also, we note that Algorithm 1 can be made conservative, that is, only adding an running the line 5 if it is a mistake. It is not hard to see that we can get the same guarantee as in Lemma A.1.

**Lemma B.2.** *Let $\mathcal{Q} = (\mathcal{X}, \mathrm{L}, \mathcal{Y}, \mathcal{C})$ be an instance of the list transductive online learning framework. Assume that* $\mathrm{D}(\mathcal{Q}) = \mathrm{d} < \infty$ *for some* $\mathrm{d}$ *in* $\mathbb{N}$. *Then, $\mathcal{Q}$ is agnostic learnable in the list transductive online learning framework.*

*Proof.* Fix $\mathrm{T} \in \mathbb{N}$. Based on the results in section 3, we know that there exists a conservative version of Algorithm 2, which can give us $\mathrm{M}^{\star}(\mathcal{Q}, \mathrm{T}) \in \mathcal{O}\left(\mathrm{L}\,\mathrm{d}\log\left(\frac{\mathrm{e}\,\mathrm{T}}{\mathrm{d}}\right)\right) \in \log(\mathrm{T})$. Let us call the best concept in $\mathcal{C}$ in the definition of regret for a given $S \in (\mathcal{X} \times \mathcal{Y})^{\mathrm{T}}$ as a sequence played by the adversary $c^{\star}$. Denote by $R^{\star}$ a sub-sequence of indices that $c^{\star}$ is correct on. Indeed, if we run the mentioned algorithm only on these point, we make at most $\mathcal{O}\left(\mathrm{L}\,\mathrm{d}\log\left(\frac{\mathrm{e}\,\mathrm{T}}{\mathrm{d}}\right)\right)$ number of mistakes on indices $J^{\star} \subseteq R^{\star}$. In fact, we only need to update our algorithm on those points. Furthermore, between all experts updating on every possible sub-sequence of size at most $\mathcal{O}\left(\mathrm{L}\,\mathrm{d}\log\left(\frac{\mathrm{e}\,\mathrm{T}}{\mathrm{d}}\right)\right)$ of $\mathrm{T}$, one of them is the one that is updating on exactly on $J^{\star}$. Thus, based on the celebrated prediction with expert advise algorithm (Cesa-Bianchi & Lugosi, 2006), we can get regret of:

$$\mathcal{O}\left(\sqrt{\mathrm{T}\,\mathrm{L}\,\mathrm{D}(\mathcal{Q})\,\log\left(\frac{\mathrm{e}\,\mathrm{T}}{\mathrm{D}(\mathcal{Q})}\right)\log\left(\frac{\mathrm{e}\mathrm{T}}{\mathrm{L}\,\mathrm{D}(\mathcal{Q})\,\log\left(\frac{\mathrm{e}\,\mathrm{T}}{\mathrm{D}(\mathcal{Q})}\right)}\right)}\right) \in o(\mathrm{T})$$

We note that our experts are at most different on $R^{\star}$ with $c^{\star}$ on $\mathcal{O}\left(\mathrm{L}\,\mathrm{d}\log\left(\frac{\mathrm{e}\,\mathrm{T}}{\mathrm{d}}\right)\right)$ number of instances. $\quad\square$

Importantly, the same technique can be used to overcome the issue of infinite label space in the work of (Moran et al., 2023), thus answering their open question.

# C. Examples

This section provides three examples of instances within the list transductive online learning framework, revealing the separations between related learnability definitions.

## C.1. L-DS Dimension

**Definition C.1** (*$i$-neighbour*). Let $f, g \in \mathscr{Y}^d$ for some non-empty set $\mathscr{Y}$ and some $d \in \mathbb{N}$. For every $i \in [d]$, we say that $f$ and $g$ are $i$-neighbours if $f_i \neq g_i$ and $\forall_{j \in [d]-\{i\}} \ f_j = g_j$.

**Definition C.2** (L-DS Dimension (Charikar & Pabbaraju, 2023)). Let $\mathcal{Q} = (\mathcal{X}, \mathrm{L}, \mathcal{Y}, \mathcal{C})$ be an instance of the list transductive online learning framework. Let $S \in \mathcal{X}^d$ be a sequence for some $d \in \mathbb{N}$. We say that $S$ is L-DS shattered by $\mathcal{C}$, if there exists $F \subseteq \mathcal{C}, |F| < \infty$ such that for all $f \in \{g \mid g \in \mathcal{Y}^d, \ \exists_{g \in F} \ \forall_{i \in [d]} \ g_i = f(S_i)\}$ and for all $i \in [d]$, $f$ has at least L number of $i$-neighbor. The L-DS dimension of $\mathcal{Q}$, denoted $\mathrm{DS}(\mathcal{Q})$, is the maximal size of a sequence $S \in \mathcal{X}^d$ for some $d \in \bar{\mathbb{N}}$ that is L-DS shattered by $\mathcal{C}$.

## C.2. Main Results on Learnability Separations

Our first result in this section implies a separation between realizable/agnostic multiclass transductive online learnability (Hanneke et al., 2024b) and realizable/agnostic list transductive online learnability.

**Proposition C.3.** *For every* $\mathrm{L} \in \mathbb{N}$*, there exists an instance of the list transductive online learning framework* $\mathcal{Q} = (\mathcal{X}, \mathrm{L}, \mathcal{Y}, \mathcal{C})$ *and another instance of the list transductive online learning framework* $\mathcal{Q}' = (\mathcal{X}, \mathrm{L}+1, \mathcal{Y}, \mathcal{C})$ *such that* $\mathrm{D}(\mathcal{Q}) = \infty$ *and* $\mathrm{B}(\mathcal{Q}') = 0$.

*Proof.* Fix $\mathrm{L} \in \mathbb{N}$. Let $\mathcal{T}$ be an infinite depth perfect rooted $(\mathrm{L}+1)$-ary tree Definition 2.9. The definition of such a tree is similar to Definition 1.7 in the work of (Bousquet et al., 2021). For every $i \in \mathbb{N}$, label all nodes at the level $i-1$ of $\mathcal{T}$ with $i-1$. Also, for every node in $\mathcal{T}$, label all its children with distinct elements from $[\mathrm{L}+1]$. Let $\mathcal{X} = \{0\} \cup \mathbb{N}$. In addition, let $\mathcal{Y} = [\mathrm{L}+1]$. Further, define $\mathcal{C} \in \mathcal{Y}^{\mathcal{X}}$ so that it only contains all functions consistent with a root-to-leaf path of $\mathcal{T}$. Now, define $\mathcal{Q} := (\mathcal{X}, \mathrm{L}, \mathcal{Y}, \mathcal{C})$. Also, define $\mathcal{Q}' := (\mathcal{X}, \mathrm{L}+1, \mathcal{Y}, \mathcal{C})$. Based on the definition of $\mathcal{T}$ and $\mathcal{Q}$, it is clear that $\mathrm{D}(\mathcal{Q}) = \infty$. Additionally, notice that for every $x \in \mathcal{X}$, the size of $\{y \mid \exists_{c \in \mathcal{C}} \ y = c(x)\}$ is bounded above by $\mathrm{L}+1$. As a result, $\mathrm{B}(\mathcal{Q}') = 0$. This finishes the proof. $\square$

Our second result in this section implies a separation between realizable/agnostic list PAC learnability (Charikar & Pabbaraju, 2023) and realizable/agnostic list transductive online learnability.

**Proposition C.4.** *For every* $\mathrm{L} \in \mathbb{N}$*, there exists an instance of the list transductive online learning framework* $\mathcal{Q} = (\mathcal{X}, \mathrm{L}, \mathcal{Y}, \mathcal{C})$ *such that* $\mathrm{D}(\mathcal{Q}) = \infty$ *and* $\mathrm{DS}(\mathcal{Q}) = 1$.

*Proof.* Fix $\mathrm{L} \in \mathbb{N}$. Let $\mathcal{T}$ be an infinite depth perfect rooted $(\mathrm{L}+1)$-ary tree Definition 2.9. The definition of such a tree is similar to Definition 1.7 in the work of (Bousquet et al., 2021). For every $i \in \mathbb{N}$, label all nodes at the level $i-1$ of $\mathcal{T}$ with $i-1$. Also, label all edges of $\mathcal{T}$ with distinct elements of $\mathbb{N}$. Let $\mathcal{X} = \{0\} \cup \mathbb{N}$. In addition, let $\mathcal{Y} = \mathbb{N}$. Further, define $\mathcal{C} \in \mathcal{Y}^{\mathcal{X}}$ such that it only contains all functions consistent with a root-to-leaf path of $\mathcal{T}$. Now, define $\mathcal{Q} := (\mathcal{X}, \mathrm{L}, \mathcal{Y}, \mathcal{C})$. Based on the definition of $\mathcal{T}$ and $\mathcal{Q}$, it is clear that $\mathrm{D}(\mathcal{Q}) = \infty$. Subsequently, we prove that $\mathrm{DS}(\mathcal{Q}) = 1$. In particular, we prove this mainly by contradiction. Assume $\mathrm{DS}(\mathcal{Q}) \geq 2$. Thus, there exist $S = (x_1, x_2) \subset \mathcal{X}$ of size 2 and $F \subseteq \mathcal{C}, |F| < \infty$ witnessing that $\mathrm{DS}(\mathcal{C}) = 2$. Without loss of generality, we assume that $x_1$ is above $x_2$ in $\mathcal{T}$. Based on the fact that the edges of $\mathcal{T}$ are labeled with distinct elements of $\mathcal{Y}$, we can conclude that every pair of concepts from $(c_1, c_2) \in \mathcal{C}^2$ such that $c_1(x_2) = c_2(x_2)$ should have be equivalent on $x_1$, meaning that $c_1(x_1) = c_2(x_1)$ as well. So, we cannot even have one neighbor. This is a contradiction. Therefore, $\mathrm{DS}(\mathcal{C}) < 2$. It is easy to see that the root node of $\mathcal{T}$ is L-DS shattered by $\mathcal{C}$. As a result, $\mathrm{DS}(\mathcal{Q}) = 1$. This finishes the proof. $\square$

Notably, one can also show that for every instance of the list transductive online learning framework $\mathcal{Q} = (\mathcal{X}, \mathrm{L}, \mathcal{Y}, \mathcal{C})$, we always have $\mathrm{DS}(\mathcal{Q}) \leq \mathrm{D}(\mathcal{Q})$.

Our third result in this section implies a separation between realizable/agnostic list online learnability (Moran et al., 2023) and realizable/agnostic list transductive online learnability.

**Proposition C.5.** *For every* $L \in \mathbb{N}$*, there exists an instance of the list transductive online learning framework* $\mathcal{Q} = (\mathcal{X}, L, \mathcal{Y}, \mathcal{C})$ *such that* $L(\mathcal{Q}) = \infty$ *and* $B(\mathcal{Q}) \leq 2$.

*Proof.* Fix $L \in \mathbb{N}$. Let $\mathcal{T}$ be an infinite depth perfect rooted $(L + 1)$-ary tree Definition 2.9. The definition of such a tree is similar to Definition 1.7 in the work of (Bousquet et al., 2021). First, label all nodes of $\mathcal{T}$ with distinct elements of $\mathbb{N}$. Also, for every node in $\mathcal{T}$, label all of its children with distinct elements from $[L + 1]$. Let $\mathcal{X} = \mathbb{N}$. In addition, let $\mathcal{Y} = \mathbb{R}$. Further, define $\mathcal{C} \in \mathcal{Y}^{\mathcal{X}}$ such that it only contains all functions consistent with a root-to-leaf path of $\mathcal{T}$ with a special property. In particular, each of these functions equals to a unique element of $\mathbb{R}$ on all instances outside its associated root-to-leaf path. Now, define $\mathcal{Q} := (\mathcal{X}, L, \mathcal{Y}, \mathcal{C})$. Based on the definition of $\mathcal{T}$ and $\mathcal{Q}$, it is clear that $L(\mathcal{Q}) = \infty$. Subsequently, we prove that $B(\mathcal{Q}) \leq 2$.

To prove this, we show that for every $T \in \mathbb{N}$, we have: $\mathbf{M}^{\star}(\mathcal{Q}, T) \leq 2$. So, we can then conclude that $B(\mathcal{Q}) \leq 2$. To see why, just notice that we can prove a lower bound based on $B(\mathcal{Q})$. In particular, if $B(\mathcal{Q}) > 2$, we can always force at least three number of mistakes to any deterministic list transductive online learning rule for large enough T.

This part of the proof is essentially identical to the similar part in the proof of Proposition 11 in the work of (Hanneke et al., 2024b). We include it for the sake of completeness. Fix $T \in \mathbb{N}$. Let $S\mathcal{X}^T$ be the sequence chosen by the adversary at the beginning of the game. Also, let $c^{\star} \in \mathcal{C}$ be the target concept chosen by the adversary. Further, let $u$ be the root-to-leaf path in $\mathcal{T}$ associated with the concept $c^{\star}$. In addition, for every $i \in [T]$, let $v_i$ be a root-to-leaf path in $\mathcal{T}$ containing first $i$ members of $S$, if it exists. Finally, let $i^{\star}$ be the smallest positive integer such that $v_{i^{\star}}$ does not exist. If $i^{\star}$ itself does not exist, let $i^{\star} = T + 1$. Our algorithm predicts according to the $[L + 1]$ labels associated with the path $v_{i^{\star}-1}$ for the first $i^{\star} - 1$ points in $S$. Moreover, if the adversary ever reveals a unique label, we use its corresponding $c \in \mathcal{C}'$ to make predictions in all future rounds. For the $i^{\star}$'th member of $S$, if it exists, we predict arbitrarily. To see that this algorithm makes at most 2 mistakes, we consider two cases. (1) If $i^{\star} = T + 1$, then our algorithm makes at most one mistake. In fact, our algorithm makes a mistake: (a) if the adversary switches the label from something in $[L + 1]$ to a unique label corresponding to the target concept $c^{\star}$. (b) perhaps on the last instance. (2) Otherwise, the algorithm makes at most two mistakes; the first mistake can be on round $i^{\star} - 1$, and the second mistake can be on round $i^{\star}$, after which the true $c^{\star}$ is known to the learner from its unique label. Indeed, if the adversary switches the label from $[L + 1]$ to a unique label corresponding to the target concept $c^{\star}$ before round $i^{\star} - 1$, we only make one mistake. In fact, we just showed that even by using an algorithm having list size of one, we can do so. This completes the proof. $\square$

Notably, one can also show that for every instance of the list transductive online learning framework $\mathcal{Q} = (\mathcal{X}, L, \mathcal{Y}, \mathcal{C})$, we always have $B(\mathcal{Q}) \leq L(\mathcal{Q})$.

