# OpenReview forum: "A Trichotomy for List Transductive Online Learning"
_ICML.cc/2025/Conference — ICML 2025 poster_

### Official Review · Reviewer_4R8d · 2025-02-17

**Overall Recommendation:** 3

**Summary:**

The paper studies the problem called list transductive online learning. That is the learner is given a sequence of instance $ (x_1,\ldots,x_T)\in\mathcal{X}$. In $ T $ round the adversary and the learner then does the following. The adversary picks an outcome $ y_{t}\in \mathcal{Y}.$ The learner is then asked to predict the outcome of $ x_{t},$ by outputing a list $ A_{t}\subseteq\mathcal{Y} $  of size $|A|= L.$ The learner announces the list $ A_{T} $.  If the learner output a list $ A $  such that $ y_{t}\not\in A_{t},$ then learner incurs an error.
    The problem is both studied in the realizable case: there exists some hypothesis class $ \mathcal{C}\subset \mathcal{Y}^{\mathcal{X}} $ such that for the sequence $ (x_{1},y_{1}),\ldots,(x_{T},y_{T}) $ there exists $ c\in \mathcal{C} $ such that for every $ i\in[T] $ $ x_{1}=c(y_{1}),$ where the algorithm performance is measured in terms of how many mistakes it makes over the $ T $ round.
    and the agnostic case, where there is no assumption on the relation between $ x_{i},y_{i} $ and $ \mathcal{C},$ and the algorithms performance is measured in terms of the regret the difference between how many mistakes the algorithm makes and how many mistakes the best hypothesis in $ \mathcal{C} $ makes.
    The paper uses $ Q=(\mathcal{X},L,\mathcal{Y},\mathcal{C}) $ as a description for an instance of a list transductive online learning problem.
    Given such a $Q$, the paper considers two complexity measures.
    The first complexity measure is the level constrained $(L+1)$-little stone dimension, the depth of the largest percept balanced $ (L+1) $ tree, with nodes labelled by instances $ x\in \mathcal{X} $, where for each level of the tree the nodes in that layer has the same value $ x $ and each of the nodes $ (L+1) $ edges connecting it to its children, are labelled with different values in $ \mathcal{Y},$ furthermore the tree is realizable by $ \mathcal{C},$ that is each root to leaf path can be realized by and hypothesis in $ \mathcal{C} .$ This is denoted $ D(Q) $
    The second complexity measure is the level constrained $(L+1)$-branching dimension, which is the largest natural number $ d $ for which there exists a perfect balanced $ L+1 $ tree(realizable by $ C $), with all the levels of the tree having nodes labelled by the same $ x\in \mathcal{X},$ and for each root to leaf path contains $ d $ instance $ x\in \mathcal{X},$ where each such instance's $ L+1$ edges connecting it to it's children are labelled with distinct values in $ \mathcal{Y}.$ This dimension is denoted $ B(Q).$
    Where the paper note that $D(Q)\leq B(Q).$
    The paper shows that the  2 complexity measures characterizes the learnability of list transductive online learning, in the realizable case.
    More specifically the paper shows that if both of the dimensions are unbounded then for any learning algorithm there exists and instance where the learner get $ \Omega(T) $- mistakes.
    If $ B(Q) $ is  unbounded, and $ D(Q) $ is  bounded then the there exists an learning algorithm making at most $ O(log(T)) $ mistakes and for any learning algorithm there exists an instance where the learning algorithm makes $ \Omega(log(T)) $-mistakes.
    If $ B(Q) $ is bounded then the learning algorithm makes $ O(1) $-mistakes.
    In the agnostic setting the paper shows that the regret is sub linear if $ D(Q) $ is  bounded, and linear if $ D(Q) $  is  unbounded.

**Claims And Evidence:**

Q: Are the claims made in the submission supported by clear and convincing evidence? If not, which claims are problematic and why?

A: The paper contains proofs for the lower bounds of the realizable case in the main of the paper. The paper claims that the remaining proofs are devoted to the appendix, which i have not checked.

**Essential References Not Discussed:**

Except for might adding a reference about where list learnability originates from i don't see any missing references.

**Experimental Designs Or Analyses:**

Q: Did you check the soundness/validity of any experimental designs or analyses? Please specify which ones, and discuss any issues.

A: I dont see how to answer the above for this theoretical article, let me know if im mistaking and give me and example how to evaluate this question, and I will do that.

**Methods And Evaluation Criteria:**

Q: Do proposed methods and/or evaluation criteria (e.g., benchmark datasets) make sense for the problem or application at hand?

A: I dont see how to answer the above for this theoretical article, let me know if im mistaking and give me and example how to evaluate this question, and I will do that.

**Other Comments Or Suggestions:**

Congratulations on your paper. The following is the notes I took will reading the article.

Line 133 namely $(x_1, x_2, . . . , x_T)$ and reveals it to the learner.

 Line 152 was Q, defined be mentioned here?

 Remark/note I really liked the open question in the end of the article discussing the possibility of list learning with infinite size - to this end could it make since to include the parameters $ L $ and $ d $ in the statement of theorem 3.1., such that the reader in general could think of how to improve the dependencies in the bound?(I would still keep theorem 1.1 informal)

 LIne 184 should it be L(Q) and not LD(Q)?

 Line 200 second column why $ -\infty $ included in N?

  Line 235-238: What about measurability in the agnostic setting? Also line 307-311: Lemma 4.1 in Cesa-Bianchi is from my understanding for finite label spaces would this course any measurability problems.

  248: Initially, an adversary chooses a sequence of T instances X $\in$ X T and reveals it to the learner.

 Definition 2.2 was $ A^* $ defined prior?

265-269 second column: Maybe adding the reference to Lemma 4.1 in (Cesa-Bianchi
    Lugosi, 2006), here also, if it also applies in this case, would help the reader.

326-329 second column: and every root
    to leaf path contains at least d nodes with edges to its children labeled by distinct elements of Y. And the tree is shattered by C.

 line 538: should it be $ V $ in the defintion of $ V_{x\rightarrow y}.$

**Other Strengths And Weaknesses:**

Strengths: I would say that the paper is well written. From my understanding of the paper being the first to study this setting of list learning it also seems original. The paper combines ideas from previous work. I enjoyed the paper. The paper also leaves an interesting question at the end of the article about whether the lists could be allowed to be unbounded, allowing for instance to output intervals of a certain size instead.

**Questions For Authors:**

1: Line 378-380 second column: How is the tree join such that realizability, nodes on same level has the same label, and is a subset of the nodes in any tree witnessing $ B(Q)=d $?

2: Agnostic and realizable: $ D(Q) $ also characterizes learnability in the realizable setting, if finite the mistake bound is sub-linear in $ T $ and if infinite linear in $ T,$(so learnable if and only $ D(Q)<\infty $ ) however the learning rates is more fine grainedly characterized depending on the finiteness of $ B(Q),$ Theorem 3.1. For the agnostic setting theorem 4.1 the statement is only about learnability which again is characterized by $ D(Q) $ bounded or unbounded. Do finiteness of $ B(Q) $ in the agnostic setting also give a more fine grained characterization of the learning rate?

3: Is it correctly understood that the algorithm 1 and 2, is not nessarily efficent since they have to consider all versionspace for labels in $ \mathcal{Y} $, which might be infinite?

Thanks for the answers to my questions. I would like to keep my score.

**Relation To Broader Scientific Literature:**

Be specific in terms of prior related findings/results/ideas/etc.
    \item A: Im not that well versed in this literature - but the cited literature seemed relevant. Specifically the paper: 1) Relates to work by for instance (Charikarm, Pabbaraju,
    2023), which studies list learning in the PAC-setup which allows, and explains how the dimension studied in that work being finite is not sufficient for characterize transductive online list learning. 2) Explains the relation to online learning(uncertainty on both labels and instances) and how that setting have been studied in list (Moran et al., 2023),  and how it previous (initiated by  (Ben-David et al., 1997)) have been studied how removing the uncertainty of the instance (so knowing the sequence in advance).(I assume that the results of (Moran et al., 2023) does not imply the results in this paper, if this is the case i encourage the authors to correct me.)
    I had one question do list learning originates from the work of Brukhim et al 2022 else I think it would be good to cite the paper originally introducing it, my apologies to the authors if I missed it.

**Theoretical Claims:**

Q: Did you check the correctness of any proofs for theoretical claims? Please specify which ones, and discuss any issues.

A: I read the proofs of the main, lower bounds for the realizable, but could have missed something while reading them. Here is my understanding of the proofs, I encourage the authors to correct my understanding. If the $ D(Q) $ dimension is infinite, then for a given algorithm $A$ the lower bound follows from there existing a $L+1$ perfectly balance tree, with depth $ T $ , realizable by $ C $, on each level the nodes are labelled by the same instance $ x\in \mathcal{X},$ and the $L+1$ edges to the nodes children have different labels. Thus when the learner is presented with the instance $ x_{i} $  on level $ i,$ and outputs a list of size $ L,$ the adversary can pick at least one branch of the tree where the learner mispredicts, since the tree have depth $ T $ the learner makes an mistake in each round. Since $ D(Q) $ is infinite, for any $ T $ such a tree can be constructed.
    In the proof of the lower bound of $ B(Q)$ being infinite, the guaranteed is that for any $ d $ there exists a realizable $ L+1 $ perfectly balanced tree, with the nodes on each level labelled by the same $ x\in \mathcal{X},$ and for any root to leaf path contains $ d $ nodes which edges to their children are label with distinct values from $\mathcal{Y}.$ Thus presenting the learner with the sequence of instance from each of the levels of the tree $ x_1,\ldots $, the adversary will again be able to make the learner make $ d $ mistakes, by on the instances where there are $ L+1 $ distinct labelled edges to children choosing the one not outputted by the learner. However since the above guarantee do not say anything about the depth of the tree and when these $d$ distinct labels occur in the sequences, it is not given that one can find a sequence of length $ T $ with this property. Thus the paper presents a way of compressing such a tree in to a subtree of depth at most $((L+1)^d-1)/L,$ while still having the above property of each root to leaf path  having $d$ nodes with distinct values on the edges to their $ L+1 $ children. Now solving $ T= ((L+1)^d-1)/L$ implies $ d= \Theta(\log(T))$(omitting dependencies on $ L $ ), thus the adversary can generate this tree of depth at most $ T $ and make the learner make $ \Omega(\log(T)) $  mistakes.

---

> ### Author Rebuttal · Authors · 2025-04-01
>
> We really thank the reviewer for dedicating their time to assess our work. In particular, we really thank the reviewer for taking the time to carefully read our paper. We are delighted that the reviewer found that our contribution is original, our paper is well written, enjoyable to read, and contains an interesting open problem. We will make sure to correct the typos and incorporate minor suggestions mentioned by the reviewer for the camera-ready version. Below, we address major comments provided by the reviewer.
>
> - Your understanding of our lower bound proofs is fully correct.
> - As you mentioned, the results of Moran et al., 2023 do not imply the results in this paper. We have an example to show this in appendix C. In fact, we mentioned that example in the last paragraph of the first column of page 2.
> - As you mentioned, list learning originates from the work of Brukhim et al., 2022.
> - Regarding measurability issues, following the work of [1], we only require a sigma-algebra so that every subset of $\mathcal{Y}$ with cardinality $\mathrm{L}$ is measurable. Notably, following the work of [1], this assumption is enough for the agnostic setting.
> - Regarding our $\log(T)$ lower bound proof, each of the trees that we want to join can witness B(Q) = d. In fact, each of them can do so even in the restricted version space based on the label of its corresponding outgoing edge of the root node. As a result, they are level-constraint (nodes of each level are assigned to a fixed instance from the instance space), and so on. Now, the main observation is that in trees witnessing the level-constraint branching dimension being equal to some k, we do not require all outgoing edges of all nodes to correspond to distinct labels. Therefore, in each of those trees, we may add a level, correspond all of its nodes to an instance from an instance space, and label all outgoing edges of it with the same label. Note that this modification still leads to a tree witnessing B(Q) = d in the restricted version space. Thus, we can make all trees have exactly similar levels by paying the number of levels less than or equal to $L \times$ maximum initial depth of them. That is why we have a factor of list size here.
> - The finiteness of B(Q) would imply a *slightly* improved upper bound in the agnostic setting. This is because, in the agnostic setting, we have $\sqrt{T}$ factor, so a logarithmic improvement that can be achieved using the finiteness of B(Q) is not a big deal.
> - Regarding the computational complexity of our algorithms, we note that our algorithms require calculating the (L + 1)-level-constrained Littlestone dimension or (L + 1)-level-constrained branching dimension for concept classes. In the special case of the binary classification, the (L + 1)-level-constrained Littlestone dimension equals the VC dimension. Moreover, we know the calculation of the VC dimension is computationally hard for general concept classes. Thus, our algorithms are not efficient for general concept classes. However, this is an issue for both PAC and adversarial online learning, even for binary classification. For instance, in the case of adversarial online learning, SOA involves computing the Littlestone dimension of concept classes defined by the online learner in the course of its interaction with the adversary, which is challenging computation, even when the concept class and the set of features are finite [2]. Notably, no efficient algorithm can achieve finite mistake bounds for general Littlestone classes [3].
>
> We hope this rebuttal has clarified your questions.
>
> Finally, once again, we thank the reviewer for taking the time to carefully read our paper and provide many helpful suggestions.
>
>
> [1] S. Hanneke, S. Moran, V. Raman, U. Subedi, A. Tewari. Multiclass Online Learning and Uniform Convergence. In Proceedings of the 36th Conference on Learning Theory, 2023.
>
> [2] P.Manurangsi, A. Rubinstein. Inapproximability of VC dimension and littlestone’s dimension. 30th Conference on Learning Theory, 2017.
>
> [3] A. Assos, I. Attias, Y. Dagan, C.Daskalakis, M. K. Fishelson. Online learning and solving infinite games with an erm oracle. 36th Conference on Learning Theory, 2023.

---

### Official Review · Reviewer_9YMd · 2025-03-14

**Overall Recommendation:** 4

**Summary:**

The authors provide theoretical analysis on the list transductive online learning problem in this paper. They first establish upper and lower bounds for the minimax number of mistakes in the realized setting, by which they solve a open problem raised from previous work. Then, in the agnostic setting, they provide an upper bound for the the minimax expected regret and solve another open problem. The key contribution in their proof is introducing two new combinatorial complexity dimensions named Level-constrained Littlestone dimension and Level-constrained Branching dimension. Finally, they raise the issue of eliminating factors from their upper bound in the realizable setting and left it for future work.

***
**Update after Rebuttal**

Thanks the authors for their responses. I encourage the authors to include the discussions about these related works in the final version. Currently, I have no other concerns. I have raised my score to 4.

**Claims And Evidence:**

All claims in this paper are theoretic results on the minimax number of mistakes in the realized setting and the minimax expected regret in the agnostic setting. They are supported by rigorous proofs.

**Essential References Not Discussed:**

From my view, I could not find related works that are essential to understanding the key contributions of the paper, but are not currently cited or discussed in the paper.

**Experimental Designs Or Analyses:**

The authors do not conduct any experiments in this work.

**Methods And Evaluation Criteria:**

The authors do not conduct any experiments in this work.

**Other Comments Or Suggestions:**

The letter $T$ in the article is in regular font in some places (Definition 2.5), while in others it is italicized (Theorem 1,1). It's better to unify them.

**Other Strengths And Weaknesses:**

The strengths of this paper is introducing novel techniques to establish theoretic bounds for the minimax number of mistakes in the realized setting and the minimax expected regret in the agnostic setting, by which two open problems are solved. The results in this paper provide learning guarantee for model in the transducitve online setting. The weakness in this paper is that the presentation needs further improvement. I encourage the authors to provide more explanations for some key concepts, such as the Level-constrained Littlestone dimension and the Level-constrained Branching dimension, by giving some examples or visualized illustrations. Besides, it would be better to place Section 1.2 after Section 2, since the readers need some background to understand the meaning of notations appear in Section 1.2.

**Questions For Authors:**

1. Could you elucidate the difference between the transductive online elarning and the traditional transductive learning introduced in [1,2,3,4,5]?
2. In addition to introducing new combinatorial complexity dimensions, is there other novelty in your proof compared with previous studies? Since you claim that two open problems are solved, why can't the previous proof techniques solve these problems, while your proof can address them?

[1] Estimation of Dependences Based on Empirical Data: Empirical Inference Science. Vladimir Vapnik,1982.

[2] Statistical Learning Theory. Vladimir Vapnik, 1998.

[3] PAC-Bayesian supervised classification: The thermodynamics of statistical learning. Olivier Catoni, 2007.

[4] Combining PAC-Bayesian and generic chaining bounds. Jean-Yves Audibert and Olivier Bousquet, JMLR 2007.

[5] Explicit learning curves for transduction and application to clustering and compression algorithms. Derbeko et al., JAIR 2004.

**Relation To Broader Scientific Literature:**

The key contribution of this work is introducing some novel combinatorial complexity dimensions named Level-constrained Littlestone dimension and Level-constrained Branching dimension, which could bring new insights for both online learning community and transductive learning community, particularly for those who working on theory. Since transducitve learning setting is also widely adopted in graph learning area, for example in the node classification task, this work could also bring some insights to graph learning community.

**Theoretical Claims:**

I roughly browsed through the entire proof process in this paper. Since I am not so familiar with the theory of online learning, I find it difficult to accurately determine whether there are issues in the proof.

---

> ### Author Rebuttal · Authors · 2025-04-01
>
> We thank the reviewer for dedicating their time to assess our work. We are delighted that the reviewer found that our paper contains novel techniques and novel combinatorial complexity measures. We will make sure to correct the typos and incorporate minor suggestions mentioned by the reviewer for the camera-ready version. Below, we address major comments provided by the reviewer.
>
> - We appreciate the reviewer's feedback regarding improvement of the presentation of our paper. We will change notations in section 2.3 to clarify the definitions in the camera-ready version. Also, we will add a figure to clarify distinction between different combinatorial structures that we have in the camera-ready version.
> - While there is a conceptual connection between transductive online learning and traditional transductive learning, as noted by Hanneke et al., 2023, there is an important distinction. Generally speaking, in online learning, we make no probabilistic assumptions regarding the data-generating mechanism. However, in traditional transductive statistical learning, we usually assume that we have an underlying data distribution, or that the learner observes labels for a uniform-random subsample of the data (as opposed to predicting online in the order given).
> - As we attempted to explain in the final paragraph of page 3, a technique inspired directly by the Halving algorithm does not yield a logarithmic  \log(T)  upper bound in our setting, even when the label space is finite. This stands in contrast to the multiclass setting, where this technique is effective when the label space is finite Hanneke et al., 2023. The main novelty in our proof of the trichotomy result lies in extending the shattering technique of Hanneke et al., 2024 to the list setting. In particular, a key and novel component of our algorithm is a new notion of shattering that exploits the sequential nature of list transductive online learning. This result appears fully in the appendix, which serves as the main technical contribution of this work. On the other hand, the solution to the second open problem follows, more or less, from a recent idea in the field.
>
> We hope this rebuttal has clarified the novelty of our contribution beyond the introduction of new combinatorial complexity dimensions.

---

### Official Review · Reviewer_yqeS · 2025-03-14

**Overall Recommendation:** 4

**Summary:**

This paper tackles the combined problem of Moran et al.'s (2023) *list online classification* and Hanneke et al's (2024) *transductive online learning* (where the sequence of instance points is given in advance). Two natural variants of Littlestone dimension are proposed combining the (L+1)-Littlestone trees of Moran et al (2023) with the level-contrained (/-branching) trees of Hanneke et al. (2024).
These two combinatorial dimensions exactly define the 3 possible mistake rates in the realizable setting (constant, logarithmic, and linear) and the first determines whether sublinear regret is possible in the agnostic setting.

**Claims And Evidence:**

Correct proofs.

**Essential References Not Discussed:**

All good.

**Experimental Designs Or Analyses:**

N/A.

**Methods And Evaluation Criteria:**

N/A.

**Other Comments Or Suggestions:**

This paper continues an interesting line of work on variants of online classification. While the proof techniques are rather standard (mostly a natural combination of Hanneke et al (2024) and Moran et al (2023)), this paper should be interesting for the theoretical ICML community. While the paper is rather incremental, I vote for acceptance.


Minor comments:

Some notation is not defined. E.g. $\Pi(..)$ presumably for distributions. Also $\mathcal{A}$ is used for subsets of labels (in the def. of $\mathcal{Y}_L$ and as the set of all deterministic algorithms.

Any reason you use $\mathfrak{s}$? Typically $\mathfrak{s}$ denotes the star number in such contexts.

Typo: A $L$-ary --> An

Consider adding an $L$ or $L+1$ index to $D(\mathcal{Q})$ and $D(\mathcal{Q})$ to make the dependence clearer (as in Moran et al (2023)? Similarly $L(\mathcal{Q})$ could be misleading as $L$ is also the list size. Why not e.g. $k$ for list size (like in some other papers on list learning)?

**Other Strengths And Weaknesses:**

Please cite Moran et al 2023 in the Def. 2.11-2.13, otherwise it might seem like you came up with these notions. Similarly maybe cite Hanneke et al (2024) for Def. 2.14, 2.15 and say that you extend their notions.

**Questions For Authors:**

Is the assumption of $|y^\star|<|x^\star|$ explicitly used anywhere? It's not a big restriction but seems unnecessary. E.g., why would you demand $y_1=y_2$ for $T=2$. Even in cases with $|y^\star|=|x^\star|$ you can obviously still "learn", e.g., if only one hypothesis is consistent to a prefix of the sequence, the learner can predict everything following the prefix correctly, even if each $x_i$ has its own new label.

**Relation To Broader Scientific Literature:**

All good.

**Theoretical Claims:**

All proofs are pretty standard and correct. See below for more comments.

---

> ### Author Rebuttal · Authors · 2025-04-01
>
> We thank the reviewer for dedicating their time to assess our work. In particular, we thank the reviewer for taking the time to verify that the work is technically correct. We are delighted that the reviewer found that our paper is interesting for the theoretical ICML community. We will make sure to correct the typos and incorporate minor suggestions mentioned by the reviewer for the camera-ready version. Below, we address a question provided by the reviewer about the assumption of $|y^{\star}| < |x^{\star}|$.
>
> - No. In fact, we use that notation to formally define Deterministic List Transductive Online Learning Rules. In particular, a Deterministic List Transductive Online Learning Rule is a mapping that maps each finite sequence of instances and a finite sequence of labels with a size *smaller* than the size of the sequence of instances to a set of size L of labels. We will change the notations in this part to clarify the definitions in the camera-ready version.

---

### Official Review · Reviewer_WvaW · 2025-03-14

**Overall Recommendation:** 4

**Summary:**

They studied the problem of list transductive online learning. In the realizable setting, they show a trichotomy of possible rates of the minimax number of mistakes.
In the agnostic setting, they show a \tilde{O}(\sqrt{T}) regret bound.

**Claims And Evidence:**

Theoretical paper, and they have proved everything theoretically.

**Essential References Not Discussed:**

I cannot think of any missing references.

**Experimental Designs Or Analyses:**

No experiments

**Methods And Evaluation Criteria:**

yes, theoretical paper

**Other Comments Or Suggestions:**

Check the questions.

**Other Strengths And Weaknesses:**

The paper is written nicely (I was confused in some parts and asked my question below), and they solved multiple open problems in earlier papers. I am not too surprised by the techniques, most of them are standard techniques in online learning literature I think (I did not go over the appendix), but the application is novel .

**Questions For Authors:**

Questions:

For the concepts in the concept class, do you assume that they give one label y_i to each example x_i or a list of labels?

Def 2.10. I am confused here.
usually in L trees each root-leaf path is labeled by a hypothsis, here each node is labeled
by a hypothesis, why is it the case here?

Def 2.11, usually in Ltree, each branch is labeled by a distinct y, here I don't think you mean that |Y|=L+1(and so all possible labeles are used in different branches), can you clarify on this?


Def 2.14, this is exactly Littlestone tree, no? Or is it possible that |Y|>L+1 and that's the difference with Littlestone tree?

Def 2.15, So a level constrained (L+1)-branching D in some sense is a subtree of a
(L+1)-Littlestone tree.

In section 2.3, I was very confused, perhaps you can add some pictures and explain which one is littlestone tree and for the ones that are not a standard littlestone tree what is the crucial difference that the reader needs to pay attention to.

Line 385-386 (the first line in this column), can you explain why (L+1) is being multiplied?

**Relation To Broader Scientific Literature:**

They study the problem of list transductive online learning. List learning studied both experimentally and theoretically. It is relevant to multi-class classification and conformal prediction.

**Theoretical Claims:**

I checked the proofs in the main body and they do make sense.

---

> ### Author Rebuttal · Authors · 2025-04-01
>
> We thank the reviewer for dedicating their time to assess our work. In particular, we thank the reviewer for taking the time to verify that our work is technically sound. We are delighted that the reviewer found that our paper contains novel applications of the techniques in the literature, and moreover mentioned that our paper is written nicely. Below, we address the questions provided by the reviewer.
>
> - We assume that each concept from the concept class assigns one label from the label space to an instance from the instance space.
> - In Definition 2.10, we just assign symbols to nodes and edges of a Perfect Rooted L-ary Tree from two abstract spaces.
> - In Definition 2.11, we just replace abstract spaces in Definition 2.10 with instance space and label space. Moreover, we may apply the property of having distinct labels corresponding to all outgoing edges of any given node in a tree in the definitions of dimensions, such as Definition 2.13. Notably, as you mentioned, we do not require to have |Y| = |L + 1|.
> - Definition 2.14 is the Definition of Level constraint (L + 1)-Littlestone dimension. Here, there are two main differences with the Definition of standard Littlestone dimension. First, the witnessing tree is (L+ 1)-ary tree instead of 2-ary tree. Second, all nodes at the same level of the witnessing tree should correspond to a single instance from the instance space.
> - In a tree witnessing the Level-constrained (L + 1)-Branching Dimension, we may have outgoing edges of a node corresponding to the same label. This contrasts a tree witnessing the (L + 1)-Littlestone Dimension that all outgoing edges of any given node should correspond to unique labels.
> - We appreciate the reviewer's feedback regarding the improvement of the presentation of our paper. We will change the notations in section 2.3 to clarify the definitions in the camera-ready version. Also, we will add a figure to clarify distinction between different combinatorial structures that we have in the camera-ready version.
> - Let us answer your question when L + 1 = 2 for simplicity. Then, the extension for an arbitrary L is straightforward. Intuitively, suppose you have two level constraint trees of the same depth. If you want to join these two trees by adding a root node and still keeping the level constraint property for the new tree, your new tree can have at most 2 times the depth of those two initial trees.

---

### Decision · Program_Chairs · 2025-05-01

**Decision:**

Accept (poster)

**Comment:**

This paper studies the theory behind a specific type of online learning called "list transductive online learning". Its main achievement is figuring out exactly how well algorithms can possibly perform in this setting. The authors introduce two new ways to measure the problem's difficulty (using special "dimensions") and show these measures perfectly predict the best possible learning rates in terms of mistakes or regret. This key result solves open questions left by earlier papers and fully characterizes the learning problem. Reviewers agreed that these theoretical results are strong and important contributions. The paper was generally considered well-written.

Overall, the assessment is positive, leaning towards acceptance. The paper is recognized for solving known problems and clearly defining the learning limits for this setting, even though the techniques might not be entirely new and the presentation could be improved for clarity.